# Reinforcement Learning for Quantum Control under Physical Constraints

Jan Ole Ernst [* 1]    Aniket Chatterjee [* 1]    Tim Franzmeyer [* 2]    Axel Kuhn [1]

## Abstract

Quantum control is concerned with the realisation of desired dynamics in quantum systems, serving as a linchpin for advancing quantum technologies and fundamental research. Analytic approaches and standard optimisation algorithms do not yield satisfactory solutions for more complex quantum systems, and especially not for real world quantum systems which are open and noisy. We devise a physics-constrained Reinforcement Learning (RL) algorithm that restricts the space of possible solutions. We incorporate priors about the desired time scales of the quantum state dynamics – as well as realistic control signal limitations – as constraints to the RL algorithm. These constraints improve solution quality and enhance computational scaleability. We evaluate our method on three broadly relevant quantum systems and incorporate real-world complications, arising from dissipation and control signal perturbations. We achieve both higher fidelities – which exceed 0.999 across all systems – and better robustness to time-dependent perturbations and experimental imperfections than previous methods. Lastly, we demonstrate that incorporating multi-step feedback can yield solutions robust even to strong perturbations. Our implementation can be found at https://github.com/jan-o-e/RL4qcWpc.

## 1. Introduction

The optimal control of quantum systems is important for enabling the development of quantum technologies such as computing, sensing, and communication, and similarly plays an important role for quantum chemistry (Brif et al., 2010) and solid state physics (Glaser et al., 2015). Quantum control requires the application of time-dependent signals

(laser pulses, microwaves etc.) to a quantum system, to realise a desired time evolution (Glaser et al., 2015; Koch, 2016; Koch et al., 2022; Mahesh et al., 2022). Examples of such tasks include system initialisation, (quantum) state preparation, gate operation, state population transfer or state measurement. Quantum control enables performing such tasks with low error rates, which is particularly important for the realisation of fault tolerant quantum computing (Terhal, 2015). Isolated quantum systems exhibit unitary dynamics (i.e. reversible) which are comparatively easy to model for modest system sizes. Yet all real quantum systems are open, subject to some interaction with the environment and require the addition of irreversible non-unitary dynamics to realistically capture their evolution (Breuer & Petruccione, 2002).

Motivated by such real-world experimental setups, we tackle quantum control with physically realistic models. The combined unitary and non-unitary evolution of quantum systems is typically modelled by a master equation (Davies, 1974; Dirr et al., 2009), a first-order linear ODE also known as a quantum Liouvillian or Lindbladian. Solving the master equation and controlling larger quantum systems is computationally expensive, growing quadratically with the quantum system size, limiting the use of standard optimisation methods. While stochastic methods (Mølmer et al., 1993) can reduce the quadratic scaling to a linear one, they require a large number of trajectories for high numerical accuracies. For large systems the only feasible option is to sample directly from a quantum device. Experimental imperfections and noise – arising from, e.g., signal distortion or attenuation in optical and electronic setups, or due to inherent system imperfections (Burkard, 2009) – pose additional challenges which existing approaches fail to address. In this work, we present a novel approach for controlling real-world open quantum systems, posing quantum control as a Reinforcement Learning (RL) problem subject to physical constraints. Specifically, we learn a control policy that maximises the fidelity of the quantum control task, while removing control signals which result in overly *fast* quantum state dynamics from the space of possible solutions. A majority of quantum control tasks, including those considered in this work, are concerned with adiabatically transferring population between quantum states (Král et al., 2007), such that the time evolution of the system is *slow* compared to the

---

[*]Equal contribution [1]Clarendon Laboratory, University of Oxford, United Kingdom [2]Department of Engineering Science, University of Oxford, United Kingdom. Correspondence to: Jan Ole Ernst <jan.ernst@physics.ox.ac.uk>.

*Proceedings of the $42^{nd}$ International Conference on Machine Learning*, Vancouver, Canada. PMLR 267, 2025. Copyright 2025 by the author(s).

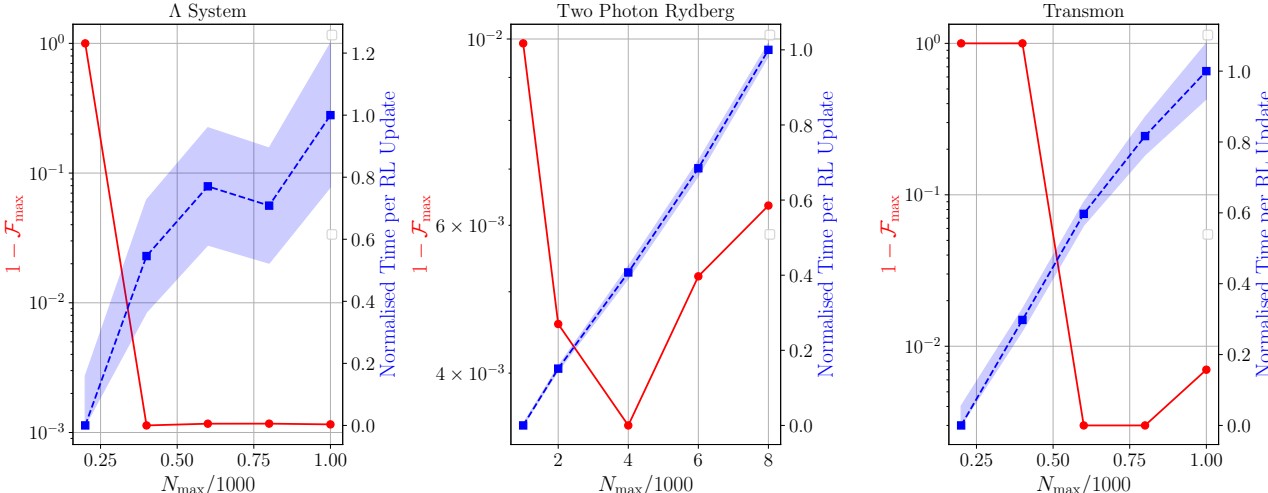

Figure 1: Minimum infidelity across hyperparameter combinations ($1 - \mathcal{F}_{\max}$, left y-axis, solid red line) and normalised GPU time per RL update (right y-axis, dotted blue line $\pm 1\sigma$) vs. permissible quantum solver steps $N_{\max}$. Limiting $N_{\max}$—which can be understood as placing an upper bound on the rate of change of the quantum system evolution induced by the control signal—improves solution quality (lower infidelity) while also increasing computational efficiency (lower normalised time).

inverse energy gap of the states ($E = \hbar\omega$) which facilitate the transfer. Quantum state dynamics which are *fast* can induce leakage errors (decay outside of the desired quantum state space). Furthermore, *fast* oscillations in the quantum state populations severely limit the robustness of control solutions to any time-dependent noise in real world experiments. In addition to the hard constraint applied to the space of possible solutions, we introduce a soft constraint that facilitates smooth pulses and fixed amplitude endpoints with finite rise-time. Both characteristics are typically required for real-world implementation of quantum control signals. Lastly, we investigate using multi-step RL to address larger levels of system noise.

Incorporating physics-based constraints into the RL problem not only enhances solution quality but also significantly improves computational scalability. Control signals inducing fast quantum state dynamics require compute-heavy simulations and thereby longer computation times. Excluding these signals enables fast parallel optimisation of multiple hyperparameter configurations by removing simulation bottlenecks.

We validate our approach on three quantum control problems. We begin with a generalised electronic $\Lambda$ system, common in quantum dots, atoms, and circuit quantum electrodynamics, revisiting a well known approach (Vitanov et al., 2017) for transferring population between states. Our implementation successfully learns realistic control signals with almost two orders of magnitude lower infidelity, and resilience to time-dependent noise. We then explore the more complex Rydberg gate (Lukin et al., 2001), crucial for

realising atomic quantum computers. Here, we demonstrate robust control signals, even in the face of noise, unlike previous approaches, and achieve higher fidelities at lower pulse energy than previous works. Lastly, we consider a superconducting Transmon quantum bit (qubit) (Egger et al., 2018b) for qubit reset, for which we discover a novel, physically-feasible reset waveform which achieves an order of magnitude higher reset fidelity than any previous work.

In conclusion, our work makes the following contributions:

1. We devise a computationally efficient RL implementation that directly incorporates physical feasibility constraints to enable discovery of experimentally realistic control signals. The improvements over standard RL algorithms are shown in Fig. 2.

2. Fig. 1 demonstrates that our constraint on the maximum number of simulation steps significantly improves computational scalability while simultaneously improving solution quality.

3. Across three quantum systems, we outperform prior methods by achieving higher fidelities, lower pulse energies, and greater robustness to time-dependent noise.

## 2. Related Work

Several algorithms exist for devising optimal time-dependent control signals for quantum systems. Analytic methods like Lyapunov (Hou et al., 2012) are effective for small isolated systems but difficult to generalise to complex environments. Gradient-based methods which consider the piece-wise evolution of a quantum system, under piece-

wise controls, such as **Gr**adient **A**scent **P**ulse **E**ngineering (GRAPE) (Khaneja et al., 2005) or Krotov (Reich et al., 2012) (which ensures monotonic convergence) work well on simple cost landscapes for a single objective with good initial guesses. Since gradients need to be evaluated exactly at each time-step for these methods, they are well suited to ideal simulations, but it is challenging to adapt them to closed-loop experimental optimisation where such gradients can rarely be determined exactly and a large number of measurements are required. RL, to the contrary, requires no intermediate quantum state information and can perform optimisation with noisy samples. Other variations of global optimal control exist which consider the optimisation of the entire time dependent control signal at once (Giannelli et al., 2022b), which are prone to local minima. Global and local optimal control methods are explored in combination in (Goerz et al., 2015). Methods which reduce the search space by decomposing the possible signal into a basis (Caneva et al., 2011) are sensitive to basis choice (Pagano et al., 2024) and evolutionary algorithms (Brown et al., 2023) lack computational scalability for larger systems or multiple objectives.

Machine learning has numerous applications in quantum science (Krenn et al., 2023). Numerous successful studies applied RL to the logical quantum ciruit level, to find optimal error correction codes (Olle et al., 2024), real time error correction (Sivak et al., 2023) or circuit complication techniques (Van Der Linde et al., 2023; Quetschlich et al., 2023). In contrast to these works, our work focusses on lower-level pulse-level control.

We review prior ML work, distinguishing between real device sampling and numerical simulations. Baum et al. (2021) devised an optimal gate set on a superconducting IBM quantum device. Reuer et al. (2023) and Porotti et al. (2022) use measurements and feedback to prepare quantum states, but generalisation to unseen environments is difficult. A model-based Hamiltonian learning approach was applied in Khalid et al. (2023), which provided important insights into sample efficiency. Although we focus on realistic quantum simulations, our RL algorithm can be readily applied to physical experiments by replacing simulations with real-device sampling.

Several studies have explored reinforcement learning (RL) for controlling simulated quantum systems. While RL has been applied to discrete action space control (Paparelle et al., 2020; An et al., 2021; Zhang et al., 2019), these methods struggle in real-world settings with analog signals exhibiting finite response time[1] and in more complex systems. We extend prior work on controlling many-body systems

---

[1]A particular limitation is the finite rise and fall time of electronic or optical signals, which refers to the time required to transition from zero to maximum amplitude (or vice versa), typically on the order of nanoseconds or greater.

(Bukov et al., 2018; Metz & Bukov, 2023; Schäfer et al., 2020) to experimentally realistic systems, incorporating control signal noise into training as suggested by Schäfer et al. (2020). While Niu et al. (2019) find time-optimal gate sequences for superconducting qubits using trust region policy-gradient methods, we advance this by enhancing computational scaleability by learning entire control signals in a single step and incorporating more complex noise models. Our control pulses for a typical $\Lambda$ system go beyond existing work (Giannelli et al., 2022a; Norambuena et al., 2023) by incorporating realistic noise models and simultaneous amplitude and frequency control to learn more optimal and realistic policies. Related work on optimising superconducting qubit gates robust to noise with RL is presented in Nam Nguyen et al. (2024), while enhanced superconducting qubit readout with RL is demonstrated in Chatterjee et al. (2025).

# 3. Background

## 3.1. Reinforcement Learning for Quantum Control

Reinforcement Learning (RL) is a framework where an agent learns to make decisions by interacting with an environment to achieve a specific goal (Sutton & Barto, 1999). In quantum control, RL can be used to find the control actions $a(t)$ that steer a quantum system toward a target state $\rho_{\text{des}}$. The key components in this RL setup are the state $s_i$, which is given as the density matrix $\rho(t)$ of the quantum system, the control action $a_i$ applied to the system, a scalar reward $r_i$, derived largely from the closeness of the system to its target state and the policy $\pi$ that maps states to actions. The objective is to learn a policy $\pi^*$ that maximises the expected cumulative expected reward, i.e. $\pi^* \in \max_\pi \mathbb{E}\left[\sum_{t=0}^{T} r_t\right]$.

**Continuous Bandit Setting** In the continuous bandit setting, the RL problem is reduced to a single-step episode. The agent selects one action $a(t)$ in a continuous space $[-1, 1]^d$, where $d$ is the action dimension representing the number of discrete action samples in time, aiming to maximise the immediate reward based on the fidelity with respect to the target state. Specifically, the optimal action $a^*$ is given as $a^* \in \arg\max_a \mathbb{E}[r(a)]$, where $r(a)$ is the reward obtained by applying action $a$.

### 3.1.1. QUANTUM DYNAMICS SIMULATION

Assessing the quality of an action $a(t)$ in quantum control involves computing the fidelity (App. (12)), which measures the overlap between the evolved quantum state and the target state. The state evolution under $a(t)$ is obtained by numerically solving the master equation (App. (11)) for $\rho(t)$, the generalised quantum state at time $t$. For a de-

tailed background on quantum dynamics and control, see App. Sec. 3.1.1. The state evolution is typically simulated using adaptive step-size solvers that implement higher-order Runge-Kutta methods (Hairer et al., 1993), which dynamically adjust their internal time steps based on local error estimates. If the error exceeds the numerical tolerance, the solver reduces its internal time step; if the error is sufficiently small, the time step is increased to enhance computational efficiency. Therefore, control signals that lead to *slower* quantum state dynamics allow the adaptive solver to use larger time steps. Hence, *slower* quantum state dynamics require fewer solver steps and less computation time.

## 4. Methods

### 4.1. Physics-Constrained Reinforcement Learning

In practice, applying RL to efficiently find high-fidelity quantum control solutions requires restricting the space of possible solutions. First, we constrain signal bandwidth, and signal area, reflecting the limited instantaneous bandwidth of electronics and signal components in experiments, and limitations in available signal power and duration. We implement these constraints via the reward function cf. Eq. 2 and Sec. 4.2, similar to the Lagrange Multiplier technique introduced in (Bhatnagar & Lakshmanan, 2012).

Simulating the quantum system to compute the reward for the RL agent at each learning step, as detailed in Sec. 3.1.1, is computationally expensive. For complex quantum systems and sub-optimal actions, this simulation can require an extremely large number of solver steps, far exceeding the computational time needed to update the RL agent. Therefore, secondly, we constrain the policy to solutions that can be simulated within a predefined number of maximum numerical solver steps, $N_{max}$. This constraint incorporates priors about the physical solution time scales into the algorithm, which incentivises adiabatic quantum state dynamics, yielding robust and interpretable solutions (cf. App. Fig. 9). Additionally, this constraint incentivises smoothness (requiring lower bandwidth) of signals, as smooth signals generally require fewer solver steps.

RL optimisation algorithms are often sensitive to the choice of hyper-parameters (Henderson et al., 2018), necessitating hyper-parameter space searching to find optimal policies. We address this challenge by synchronously optimising control policies for up to 1024 RL agents in parallel on a single GPU device, by implementing both the quantum solver and the RL algorithm using JAX (Bradbury et al., 2018), which features just-in-time compilation and automatic differentiation. This allows for the compilation of the parallelised training and simulation loop end-to-end. However, in this synchronised parallel setup, the quantum simulation time needed per step is governed by the maximum quantum simu-

lation time across all hyper-parameter configurations, as the slowest simulation among all learned policies determines the speed of the entire loop. We mitigate this bottleneck with a constrained RL algorithm that solves the quantum control problem subject to the condition that the required number of quantum simulation steps does not exceed a chosen threshold $N_{max}$. Formally, this constrained RL problem, for which the quantum simulation can be executed in fewer than $N_{max}$ steps, is defined as:

$$\pi^* \in \max_\pi \ \mathbb{E}\left[\sum_{t=0}^{T} r_t\right] \ \Big| \ \text{for } a \in \pi \ \ N_{\text{Sim}}(a) < N_{\text{Sim}}^{\max} \quad (1)$$

where $\pi$ is the policy, $r_t$ is the reward at time $t$, and $N_{\text{Sim}}(a)$ is the number of solver steps required for conducting the quantum simulation for an action $a$ sampled from policy $\pi$. Implementing this constrained RL algorithm prevents bottlenecks as it ensures that all simulations within the vectorised run are completed within a fixed maximal time frame. This approach enables efficient hyper-parameter exploration with minimal computational cost. The constraint, though seemingly restrictive, is physically motivated for adiabatic population transfer between quantum states (Král et al., 2007). In adiabatic processes, the system evolves slowly relative to the inverse energy gap, requiring fewer solver steps. The maximal effective Rabi frequency, $\Omega_{\text{eff}} = \frac{\overline{\Omega}^2}{\overline{\Delta}}$, sets a lower bound for $N_{max}$ via the adiabaticity condition $\Omega_{\text{eff}} \cdot \delta_t \gg 1$ (Král et al., 2007). In practice, we increment $N_{max}$ as long a significant decrease in infidelity is observed (see Fig. 1 for infidelities at different maximum solver steps for different quantum systems).

To summarise, the constrained RL approach not only improves computational efficiency but also promotes the selection of physically realistic control signals (cf. App. Fig. 14). By enforcing these constraints, we achieve more interpretable quantum state dynamics (cf. App. Fig. 9) and prioritise adiabatic solutions, leading to experimentally feasible, robust and high fidelity control techniques.

### 4.2. Reward Shaping

We parametrise the control signal as a combination of time-dependent amplitudes $\Omega_i$ and time-dependent frequencies $\Delta_i$ and introduce smoothness constraints that facilitate efficient learning and further improve computational efficiency. Smoother waveforms are easier to implement experimentally, offer clearer interpretation of the optimal quantum state evolution, and significantly speed up simulation times by reducing the number of required solver steps. To facilitate smooth signal discovery, we apply a Gaussian convolution filter to our control signal with a standard deviation $t_\sigma$ (cf. App. (20)) before simulating the quantum state dynamics which improves learning dynamics by favouring slower solution dynamics (an ablation over this is found in App.

Sec. G.4 Fig. 13). The reward function contains auxiliary smoothing penalties and is defined as

$$L_a = -w_{\mathcal{F}} \log\left(1 - \mathcal{F}\right) - w_\Omega \text{ReLU}\left(\frac{\sum S(\Omega_i)}{\sum S_{\text{base}}} - 1\right)$$
$$- w_\Delta \text{ReLU}\left(\frac{\sum S(\Delta_i)}{\sum S_{\text{base}}} - 1\right) - w_A \frac{\sum A(\Omega_i)}{A_{\text{base}}} \quad (2)$$

The first and most important reward-function term incentives high fidelity $\mathcal{F} = \mathcal{F}(\rho_{\text{fin}}, \rho_{\text{des}})$ with respect to the desired final state $\rho_{\text{des}}$. This fidelity reward scales with the logarithm infidelity $\log\left(1 - \mathcal{F}\right)$. Next we define smoothness penalties, where $\text{ReLu}(x)$ defines the ReLu function: $\text{ReLu}(x) = 0$ if $x < 0$ || $\text{ReLu}(x) = x$ if $x >= 0$. The smoothness $S$ is compared to that of a reference signal $S_{\text{base}}$ (cf. App. Sec.G.4 Fig. 15 for a definition of and ablation over different smoothing functions). We introduce a smoothness penalty weighted by small coefficients $w_\Delta, w_\Omega$, to balance fidelity, interpretability, and computational efficiency. Fig. 13 shows an ablation over various smoothing penalties. Contrary to the $\Lambda$ system and Rydberg atom, the Transmon favours stronger smoothing penalties showing that our approach is adaptable to a wide variety of physical problem settings. Larger values of $t_\sigma$ and $w_\Delta, w_\Omega$ also reduce the maximum required solver steps and thereby further enhances computational scaleability. The ability to achieve high-fidelity solutions across all environments at larger convolution standard deviations, $t_\sigma$, also demonstrates that we can find optimal signals compatible with realistic electronic control systems with limited instantaneous bandwidth (cf. App. Fig. 14).

The final reward term penalises solutions with large pulse area (cf. App. Sec. C Fig. 7 for an ablation over different area penalties for the $\Lambda$ system), we set $w_A = 0$ for the Rydberg and Transmon problem settings. We introduce additional physics-informed constraints which are problem specific and defined in App. Sec. C and App. Sec. D.

# 5. Experiments

**Overview**   We primarily focus on the continuous bandit RL setting, with supplementary experiments showing multi-step RL outperforms bandit methods under strong signal perturbations. We conduct experiments on three critical quantum control tasks relevant to quantum information processing. First, we address population transfer in multi-level $\Lambda$ systems, relevant to quantum chemistry and solid state physics, where we achieve high-fidelity population transfer in spite of dissipation and cross-talk. We explicitly show the effect of our physics driven constraints and the superiority of PPO over other RL alternatives in Fig. 2. Secondly, we optimise Rydberg gates in neutral atom quantum devices, focusing on enhancing gate fidelities and robustness to time-dependent noise, which is crucial for scalable quantum computing. Thirdly, we develop efficient reset protocols

for superconducting Transmon qubits under bandwidth constraints, essential for fast quantum circuit execution and scaling the volume of quantum gates that can be performed. Here, we discover a novel, physically-feasible reset waveform which achieves an order of magnitude higher reset fidelity than any previous work. Fig. 1 demonstrates the efficacy of our proposed method in finding higher-fidelity solutions while reducing computational demand. Although in this work we simulate realistic and noisy quantum systems, we detail in App. Sec. F how our setup can be applied to real systems directly.

### 5.0.1. EXPERIMENTAL IMPLEMENTATION

**Constrained RL Implementation**   To enforce the constraint $N_{\text{Sim}}(a) < N_{\text{Sim}}^{\text{max}}$ on actions $a$ sampled by policies $\pi$ in the bandit setting, we assign a penalty reward $r_{\text{penalty}}$. In the bandit setting, $r_{\text{penalty}}$ is assigned to any policy where $N_{\text{Sim}}(a) >= N_{\text{Sim}}^{\text{max}}$, and the value is chosen to be lower than any other possible reward in the environment, ensuring that the optimal policy cannot include states violating the constraint (Altman, 2021). This approach can be easily extended to multi-step settings when the bounds of the reward function are known, which is the case here (Altman, 2021). The final reward function is then defined as

$$L = \begin{cases} r_{\text{penalty}} & \text{if } N_{\text{Sim}}(a) >= N_{\text{Sim}}^{\text{max}} \\ L_a & \text{else} \end{cases} \quad (3)$$

where $L_a$ is defined in (2). Further details on the implementation, including all relevant libraries, is found in App. Sec. G.1. All the code is fully open source at Ref. Ernst et al. (2025)

### 5.1. Population Transfer in Multi-level $\Lambda$ System

Controlling quantum dynamics in multilevel systems is crucial for quantum information processing and relevant to solid-state physics and chemistry (Bergmann et al., 2019; Vitanov et al., 2017). We study a common experimental setup (Vitanov et al., 2017), the $\Lambda$ system, featuring multiple ground and excited states, with the latter subject to spontaneous decay. We consider two time-dependent control signals with amplitudes $\Omega_S, \Omega_P$ which couple two electronic states with relative time-dependent frequency detunings $\Delta_P$ and $\Delta_\delta$ (cf. App. Sec. C for more details). These four parameters define the control fields in (10). While analytically optimal pulses exist for idealised three-level systems (Kuklinski et al., 1989; Vasilev et al., 2009), we include excited state dissipation, parametrised by rate $\Gamma$, and an additional excited state detuned positively by $\Delta_X$ requiring cross-talk suppression (cf. App. Sec. C for details). This represents a common physical configuration, describing nitrogen vacancy centres (Balasubramanian et al., 2009), quantum dots (Economou et al., 2012), and single atoms (Ernst et al., 2023). We present and benchmark results on optimising

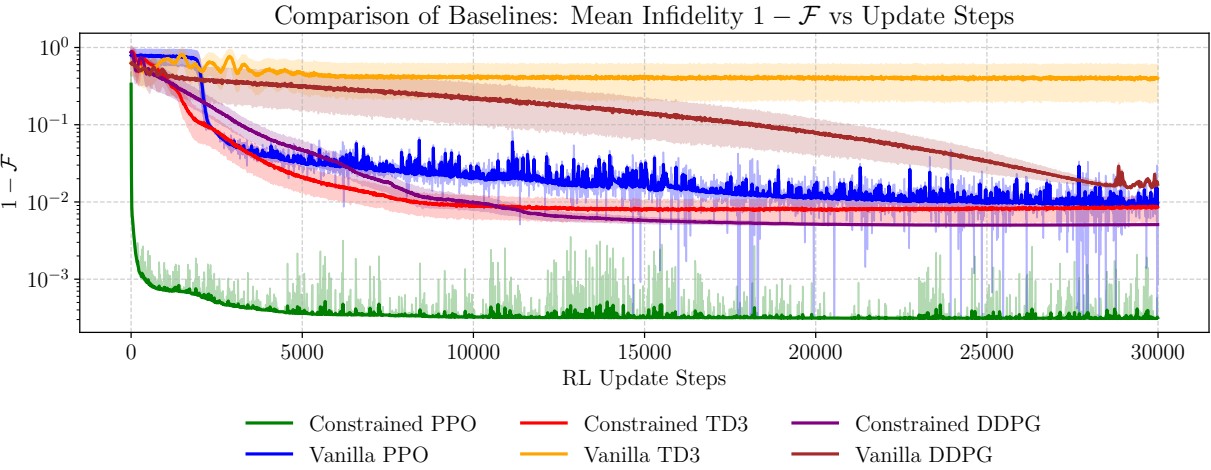

Figure 2: We compare three model-free RL algorithms — PPO (Schulman et al., 2017), DDPG (Lillicrap et al., 2016) and TD3 (Fujimoto et al., 2018) in the noise-free Λ-system setting. Each algorithm is run on the same Nvidia P100 GPU, using a single set of optimised hyper-parameters per algorithm, with a common batch size and results averaged over two seeds (one s.d. shaded) for the same number of total steps. We compare standard (vanilla) versions of the algorithms which use $L = \mathcal{F}$, to our constrained formulation, using the reward defined in Eq. 2 and incorporating step-size penalties (cf. Eq. 3). The constrained variants consistently outperform their vanilla counterparts, demonstrating the effectiveness of the constrained problem formulation and physically motivated loss function. Our implementation of PPO achieves the best performance, surpassing DDPG and TD3 by over an order of magnitude in mean fidelity. Furthermore, PPO reaches mean fidelities exceeding 0.99 up to $100\times$ faster than the alternative RL approaches.

population transfer from one ground state to another, fixing $\Gamma = 1$, $\Omega_{max} = 30$ and $\Delta_X = 100$.

We observe in Tab. 1 that the fidelities $\mathcal{F}$ achieved in a 4-level Λ system are significantly higher than state of the art and also more robust across different random initial seeds, highlighting the superiority of RL over methods which directly differentiate the control action with respect to the fidelity. We further find that the learned pulses are physically viable, while prior work (Giannelli et al., 2022b; Brown et al., 2021) found infeasible solutions, which exhibit non-zero amplitudes at the start or end or have instantaneous parameter changes which cannot be realised on bandwidth limited hardware. Moreover, we note that our implementation of PPO achieves the best fidelity $> 0.999$ and also achieves $> 0.99$ fidelity 100x times faster than TD3 (Fujimoto et al., 2018) or DDPG (Lillicrap et al., 2016) as shown in Fig. 2. Sweeping $w_A$ cf. (3) we find signals which have pulse extremely low pulse areas which approach the lower bound quoted in (Norambuena et al., 2023) (cf. Fig. 7 in the App.), implying low signal energy requirements. Example signals differ significantly for different pulse area penalties which is shown in Fig. 3.

Random fluctuations or noise of either signal $\Omega_{S/P}$ or $\Delta_{\delta/P}$ are not detrimental to the overall fidelity. We implement an Ornstein–Uhlenbeck noise process for both $\Delta_{\delta/P}$ and $\Omega_{S/P}$, a noise model which creates continuous noise $\nu_t$ in

time with mean $\mu$ and standard deviation $\sigma$ (for details cf. App. Sec. G.5). Such noise typically arises from a variety of imperfections in the signal chain, as well as quantum system level noise, such as magnetic field fluctuation or motion. Using unbiased ($\mu = 0$) noise with various standard deviations exemplifies good robustness to low noise levels as shown in Fig 8 (cf. App. Sec. C) where we attain $> 0.99$ mean fidelity for $\sigma_\Omega = \sigma_\Delta = 0.1$. Further increasing $\sigma$ leads to significantly reduced population transfer fidelities which we address with multi-step RL in Sec. 5.4. Solutions for a larger variety of system parameters and an extension to partial state transfer are shown in App. Sec. C.

### 5.2. Rydberg Gates

Neutral atom quantum devices have shown promise for realising scalable, logical quantum computing (Bluvstein et al., 2023). The realisation of quantum computing requires a two-qubit gate (Nielsen & Chuang, 2010) which relies on the interaction of multiple atomic qubits which are brought in relative proximity (a detailed description of the Hamiltonian is provided in App. Sec. D) and addressed with laser beams. We consider an optimisation of the Rydberg gate (Lukin et al., 2001) under realistic experimental conditions and signal perturbations.

We consider the most widespread implementation of a Rydberg $C$-$Z$ gate (a single photon Rydberg gate (Levine et al.,

| Method | $\mathcal{F}$ | Notes |
|--------|---------------|-------|
| Optimal Control (Giannelli et al., 2022b) | $\overline{0.890} \pm 0.064$ | Using BFGS (Fletcher, 1987), n_iter = 1000. |
| Krotov (Goerz et al., 2019) | $\overline{0.99} \pm 0.001$ | n_iter=10000 |
| Analytic (Vasilev et al., 2009) | $0.901$ | No uncertainty as analytic solution. |
| Reinforce (Brown et al., 2021) | $\overline{0.930} \pm 0.034$ | Not exp. feasible |
| PINN (Norambuena et al., 2023) | $0.83$ | No code available to benchmark |
| Vanilla DDPG (Lillicrap et al., 2016) | $\overline{0.985} \pm 0.004$ | Vanilla implies reward $L = \mathcal{F}$ |
| Vanilla TD3 (Fujimoto et al., 2018) | $\overline{0.625} \pm 0.01$ | – |
| Vanilla PPO (Schulman et al., 2017) | $\overline{0.989} \pm 0.0002$ | – |
| PPO (this work) | $\mathbf{0.999 \pm 0.0003}$ | – |

Table 1: We benchmark different methods for optimising coherent quantum population transfer in a multilevel $\Lambda$ system. Averaged over 32 random seeds, our method achieves significantly higher $\mathcal{F}$ than prior work with reduced sensitivity to the initial seed, yielding experimentally feasible controls. Timing comparison is provided in App. Sec. G.3.

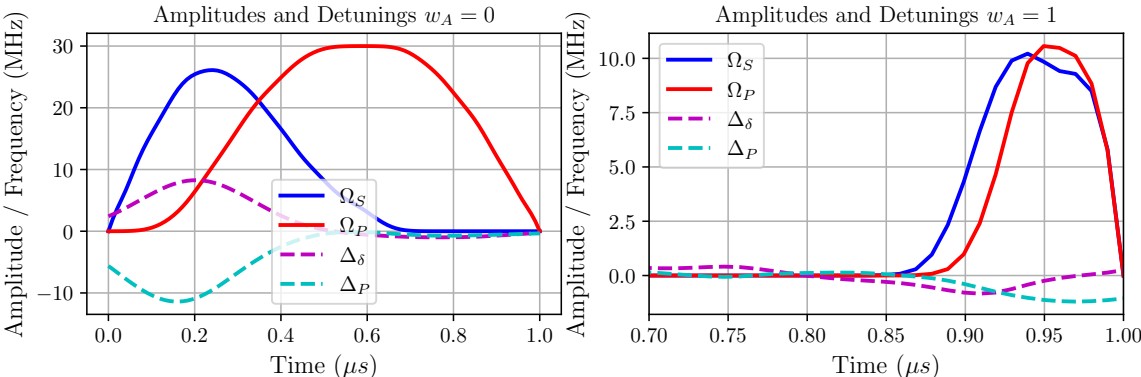

Figure 3: Shown are example control signals generated for different pulse area penalties. For $w_A = 0$ (left), the algorithm seeks to maximise $\Omega$ at all times after a fast rise and compensates cross-talk with frequency chirping. For $w_A = 1$ (right), we plot only the time interval $[0.7, 1]$, as the pulse amplitudes are zero otherwise. We show that the we discover pulses which reminisce of two interleaved Gaussians, but exhibit non zero two-photon detuning $\Delta_\delta = 0$ to cancel cross-talk (cf. App. Sec. C), which differs from the original experimental proposal for coherent population transfer (Vitanov et al., 2017).

2019; Jandura & Pupillo, 2022) with a single pulse of amplitude $\Omega_P$ and time-dependent frequency $\Delta_P$ which has known solutions. This is compared to the two-photon Rydberg $C$-$Z$ gate which uses two time-dependent signals with amplitudes $\Omega_P, \Omega_S$ and frequencies $\Delta_P, \Delta_S$ (akin to the $\Lambda$ system). The single photon Rydberg gate is vulnerable to time-dependent noise which motivates the determination of an optimal pulse sequence for the two-photon Rydberg gate which exhibits superior robustness to external perturbations (cf. App D). Finding optimal protocols which simultaneously optimise both amplitude and frequency of Pump and Stokes beams is challenging since there exists a large number of possible control signals in a large Hilbert space. A similar setup was addressed in Goerz et al. (2014), however this did not explicitly consider the decay of the Rydberg state and the optimised signals are extremely challenging to realise experimentally. Compared to Saffman et al. (2020) we find a solution (cf. App. Fig. 10) which is higher fi-

delity $\mathcal{F} = 0.9996$ than the analytic solution $\mathcal{F} = 0.99$, as well as the numerical solution $\mathcal{F} = 0.997$ and faster $0.25\mu s$ compared to the $1\mu s$ numerical solution. What is also remarkable is that for moderate levels of unbiased time-dependent amplitude and frequency noise (cf. App. Sec. D) we observe $\overline{\mathcal{F}} > 0.999$. Compared to Sun (2023), who neglect intermediate state decay, we achieve similar fidelities but with an order of magnitude lower peak Rabi frequencies which implies lower laser power requirements. Moreover, we implement a direct C-Z gate which does not require any additional single qubit rotations. We directly differentiated the input action with respect to the fidelity with a BFGS (Fletcher, 1987) method over 1000 iterations and for 32 random initial seeds and achieved a mean fidelity of $0.914 \pm 0.0742$ (one s.d.) showing the superiority of RL to reliably achieve high fidelity solutions. The enhanced computational scaleability offered by our implementation could be used to optimise multi qubit gates with $k > 2$

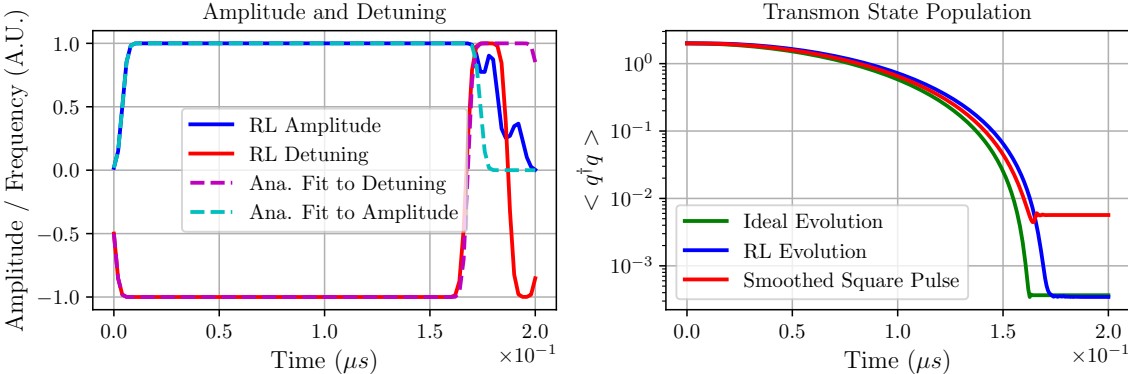

Figure 4: Optimal Waveform for Transmon Reset (left) discovered by RL and corresponding state evolution (right). The RL waveform (solid lines) amplitude evolution reminisces of a square-top Gaussian, with a smooth Heaviside-detuning that accounts for time-dependent frequency shifts. Equivalent reset performance is found by fitting a Heaviside-detuning reset and a Gaussian square amplitude waveform (dashed lines), simplifying experimental calibration. Our approach shows reset errors of 0.03% matching the performance under an experimentally unrealistic ideal square pulse, and showing an order of magnitude improvement over a smoothed square pulse.

which are also robust.

### 5.3. Transmon Reset

Superconducting quantum bits (qubits) have played a central role in quantum computing breakthroughs, including the demonstration of quantum supremacy (Arute et al., 2019) as well as the suppression of errors with the surface code (Acharya et al., 2023). The Transmon (Koch et al., 2007), a widely used superconducting qubit, operates within its two lowest energy levels to form a qubit subspace. Recent advances have extended Transmon lifetimes beyond 0.5 ms (Wang et al., 2022), enabling longer quantum circuits and the implementation of error correction codes. To maximise circuit operations within the qubit's lifetime, Transmons must be reset efficiently with high fidelity.

Two main reset techniques exist: conditional reset (Ristè et al., 2012), which follows state measurement, and unconditional reset (Magnard et al., 2018), which is faster and more robust. We focus on optimising waveforms for unconditional reset (cf. App. Sec. E for further details). The reset rate is proportional to drive strength, theoretically favouring high-amplitude square pulses for maximum fidelity. However, a drive-induced Stark shift alters the Transmon's resonance frequencies (Zeytinoğlu et al., 2015). In ideal conditions, a square pulse with a calibrated frequency can counter this shift. IBM demonstrated this approach experimentally, achieving 0.983 fidelity, while simulations under ideal conditions reached 0.996 fidelity (Egger et al., 2018a). This mismatch could be explained by experimentally realistic bandwidth constraints as square pulses have a finite rise and fall time, which induces a time-dependent

frequency shift. While optimal control (Gautier et al., 2024) has been applied to the task of reset pulse optimisation, minimal bandwidth constraints implied that no novel waveforms were found for improving the reset transition in a simple and realistic experiment. Using BFGS with direct differentiation of the input signal failed to optimise multi-objective reward functions or satisfy realistic signal constraints. When optimising solely for fidelity, it remained slow and prone to local minima due to the large search space of non-smooth actions. Similarly, we were unable to attain any reasonable learning with alternative RL algorithms other than PPO.

We apply our RL method to optimise the Transmon reset waveform under bandwidth constraints imposed by Gaussian-smoothing (for further details cf. App. Sec. G.4). Considering state of the art parameters, as given in the IBMQ experiment – a qubit lifetime $T_1$ of $500\mu s$ – we find that our RL approach achieves 0.9997 fidelity under realistic bandwidth constraints shown in Fig. 4 (cf. App. E for further implementation details). This is compared with a perfect square pulse - which is not experimentally realistic - without any smoothing, and a calibrated square pulse with smoothing - which represents prior work (Egger et al., 2018a). The RL waveform matches the theoretical optimal fidelity of the perfect square pulse and improves the fidelity of waveform used in prior work (Egger et al., 2018a) by an order of magnitude. In App. Sec. E we explicitly compare the results with the parameters used in Egger et al. (2018a), and find that the RL-discovered reset waveform achieves the fidelity 0.997 of the ideal square pulse compared to the measured fidelity of 0.983. A fitted Heaviside detuning function from the RL-discovered waveform corrects the drive-induced Stark shift, simplifying experimental calibra-

tion, which we dub Heaviside-Corrected Gaussian Square (HCGS) and explain further in App. Sec. E.1. Further results and extensions are provided in App. Sec. E.

### 5.4. Multi-Step Reinforcement Learning

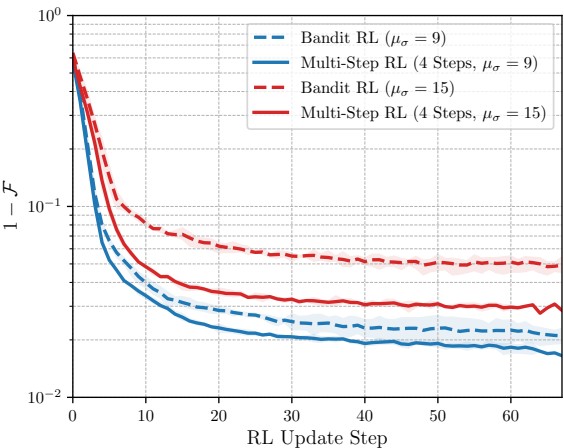

Figure 5: Comparison of mean infidelity for different values of $\mu_\sigma$ for bandit RL and multi-step RL. We observed a 2x reduction in infidelity for larger noise bias $\mu_\sigma$ by using multi-step RL over the bandit setting.

We study the effectiveness of multi-step reinforcement learning strategies in achieving high fidelity control solutions under adverse noise conditions. Feedback on nanosecond timescales has been demonstrated experimentally (Álvarez et al., 2022; Koch et al., 2010), supporting this approach. This feedback can be realised by measuring classical signal noise without affecting quantum coherence. For example, in atomic quantum systems, laser intensity $I$ can be monitored separately from the quantum system, as $I \propto |\Omega|^2$. Changes in I directly modulate $\Omega_{S/P}$ and thereby provide feedback.

In multi-step RL, the agent aims to maximise cumulative rewards over multiple steps, unlike the bandit setting where actions are independent. In our setup, at the start of each episode, a parameter $\mu_\Omega$ is sampled uniformly from $[-\mu_\sigma, \mu_\sigma]$ to initialise an Ornstein–Uhlenbeck noise process (see App. Sec. G.5 (25)). The agent's control signal $a_t = \Omega_i(t)$ (amplitudes only) is affected by this noise, resulting in $\Omega_i' = \Omega_i + \nu_t$. In bandit RL, the agent does not observe the noise $\nu_t$ and selects the action in one step. Conversely, in multi-step RL, each episode is divided into four sections of 8 action samples corresponding to $0.25\mu s$ each. The agent initially observes $O_t = 0$ but receives the value of $\mu_\Omega$ at times $t = 0.25$, $0.5$, and $0.75\mu s$ (further implementation details are given in App. Sec. A.2). Fig. 5 illustrates that multi-step RL outperforms the bandit approach, especially as $\mu_\sigma$ increases beyond 10.

## 6. Conclusion

In this work, we introduced a novel reinforcement learning implementation for controlling open quantum systems by formulating quantum control as a constrained RL problem. By integrating physics-based constraints that exclude control signals inducing overly fast quantum dynamics and enforcing smooth pulses with finite rise-time, we enhanced both the quality of control solutions and computational scalability. Our approach outperformed existing RL and non-RL methods on three key quantum control tasks, achieving higher fidelities and increased robustness to time-dependent noise. Achieving $> 0.999$ fidelity for the environments is significant insofar as this is often quoted as the threshold for error free quantum computing with error correction (Gottesman, 2002; Fowler et al., 2012). We wish to highlight here, that especially for the Transmon qubit, we find novel waveforms that can be described with smooth functional parametrisation and realised with standard hardware. We are actively working on verifying the quality of our found solutions on physical devices. For future work, we envision extending our approach to more complex quantum systems as required for fault tolerant quantum computing with error correction, this includes multi-qubit systems and higher-dimensional state spaces. Exploring adaptive constraint mechanisms that adjust during the learning process could further improve performance in these large systems. Additionally, future work would extend this to quantum control tasks which require multiple sequential quantum gates or other concatenated control operations. Developing generalised quantum control policies that incorporate system-dependent observations during learning would enable a single policy to adapt across diverse qubits and devices, significantly enhancing scalability. Validating these solutions on real quantum hardware would accelerate the practical advancement of quantum technologies. We also wish to highlight that the methods presented here can address a variety of complex control tasks in real-world physical systems, even in the presence of noise, imperfections, and parameter drifts.

**Limitations** While our physics-constrained RL implementation enhances computational efficiency and solution quality, it may limit the exploration of control strategies that involve very fast and non-adiabatic quantum dynamics. The method's effectiveness also relies on accurate modelling of quantum systems, so models would first have to be established for black box systems or more complicated real world devices. Although we address certain types of noise and perturbations, fully accounting for all experimental imperfections is an area for future work and we could consider grey-box models of real devices.

## Acknowledgements

JOE is grateful for helpful discussions with Daniel Puzzuoli and Ronan Gautier on GPU based quantum simulations and the authors are grateful to Christiane Koch for providing valuable feedback on this manuscript.

The authors would also like to acknowledge the use of the University of Oxford Advanced Research Computing (ARC) facility in carrying out this work (Richards, 2015). JOE & AK acknowledge financial support provided by the Christ Church Research Centre and the EPSRC funded QCI3 Quantum Hub.

## Impact Statement

This paper presents work whose goal is to advance the field of quantum control by applying machine learning techniques. We devise novel, higher fidelity, and more robust time dependent control signals for existing quantum systems. This work is unlikely to directly and significantly influence the applications of quantum technologies in the near term. However, in the long run, it may contribute to the development of advanced quantum technologies, which can have wide ranging impacts.

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

# Appendix

Here we present detailed explanations of the extended RL background, quantum dynamical systems simulated in the main paper, show auxiliary results and explain our implementation in greater detail.

## A. RL Background

### A.1. Bandit Setting in Reinforcement Learning

In the bandit setting, the RL problem is simplified as there is no state transition, only actions and rewards. Each action $a \in \mathcal{A}$, which are time-dependent quantum control signals $\Delta_i, \Omega_i$ yields a reward from a stationary probability distribution. The objective is to maximise the expected reward over a sequence of actions.

Formally, given a set of actions $\mathcal{A}$, each action $a \in \mathcal{A}$ has an unknown reward distribution with expected reward $R(a)$. The goal is to find the action $a^*$ that maximises the expected reward:

$$a^* = \arg\max_{a \in \mathcal{A}} \mathbb{E}\left[R(a)\right] \qquad (4)$$

This setting forms the basis for more complex RL problems.

### A.2. Extended Time Horizon in Multi-step RL

For multi-step RL, we consider an extended time horizon. In contrast to the bandit setting, each episode is divided into four sections, each of $1/4$ the length of the total number of action samples comprising the control signal. The agent does not observe any information about the noise at time step $t = 0$, with the observation $O_t = 0$. However, at time steps $t = 0.25, 0.5$, and $0.75$, the agent receives the value of mean noise $\mu_\Omega$ sampled at the beginning of the episode. Formally, the observation function $\mathcal{O}_t$ is defined as:

$$\mathcal{O}_t = \begin{cases} 0 & \text{if } t = 0 \\ \mu_\Omega & \text{if } t = 0.25k \text{ for } k = 1, 2, 3 \end{cases} \qquad (5)$$

The agent's policy $\pi$ then uses this observation to decide the action at each time step, where $\tilde{S}_t$ is the union of the state in the bandit setting $s_t$ and the observation $\mathcal{O}_t$ which defines an action $a_t$ through a conditional probability distribution $\mathcal{P}$:

$$\pi(\tilde{s}_t, a_t) = \mathcal{P}[a_t | \tilde{s}_t, \theta] \qquad (6)$$

In general extended time horizon RL, the agent must consider the long-term consequences of its actions. This is formalised through the discount factor $\gamma$, which ensures that future rewards are appropriately weighted. Given that we have a fixed number of four steps we set the discount factor to zero.

### A.3. Proximal Policy Optimisation (PPO)

Proximal Policy Optimisation (PPO) (Schulman et al., 2017) is a popular algorithm in modern RL, combining the benefits of policy gradient methods with stability improvements. PPO aims to optimise the policy by ensuring that updates do not deviate too much from the previous policy. This is achieved using a clipped objective function.

The objective function in PPO is defined as:

$$L^{\text{CLIP}}(\theta) = \mathbb{E}_t\left[\min\left(r_t(\theta)\hat{A}_t, \text{clip}(r_t(\theta), 1 \pm \epsilon)\hat{A}_t\right)\right] \quad (7)$$

where:

- $r_t(\theta) = \frac{\pi_\theta(a_t|s_t)}{\pi_{\theta_{\text{old}}}(a_t|s_t)}$ is the probability ratio under the new and old policies.
- $\hat{A}_t$ is an estimate of the advantage function at time-step $t$.
- $\epsilon$ is a hyper-parameter that controls the clipping range.

The clipping mechanism in the objective function ensures that the new policy does not deviate significantly from the old policy, thereby improving training stability and preventing large, destabilising updates.

PPO also incorporates an entropy bonus to encourage exploration and prevent premature convergence to suboptimal policies. The overall objective with the entropy bonus can be written as:

$$L(\theta) = \mathbb{E}_t\left[L^{\text{CLIP}}(\theta) + c_1\hat{A}_t + c_2 E[\pi_\theta](s_t)\right] \qquad (8)$$

where $c_1$ and $c_2$ are coefficients, and $E[\pi_\theta](s_t)$ denotes the entropy of the policy at state $s_t$.

In summary, PPO effectively balances exploration and exploitation while ensuring stable policy updates, making it a robust choice for RL in quantum control tasks.

## B. Quantum Dynamics & Control

Quantum dynamics describes the time evolution of quantum systems. A system's state is represented by a *quantum state*, a vector in a complex Hilbert space $\mathcal{H}$. The most common representation is the *state vector* $|\psi\rangle \in \mathcal{H}$. A pure quantum state is described by a normalised vector (Nielsen & Chuang, 2010) $|\psi\rangle = \left(\psi_1, \quad \psi_2, \quad \cdots \quad \psi_n\right)^\top$, where $\langle\psi|\psi\rangle = 1$. A more general representation is the *density matrix* $\rho$, which for a pure state is $\rho = |\psi\rangle\langle\psi|$, (Nielsen & Chuang, 2010), and extends to classical mixtures of pure quantum states. The quantum state populations are defined as $|\psi_i|^2$ (i.e. the diagonal terms of $\rho$). Operators in quantum mechanics are unitary, making dynamics reversible. The unitary time evolution of $|\psi(t)\rangle$ is governed by the *time-dependent Schrödinger equation*:

$$i\hbar \frac{\partial}{\partial t} |\psi(t)\rangle = \hat{H} |\psi(t)\rangle , \qquad (9)$$

where $\hbar$ is the reduced Planck constant, and $\hat{H}$ is the Hamiltonian operator representing the system's total energy. Quantum control manipulates systems to achieve desired dynamics using time-dependent control fields, represented by the *control Hamiltonian*. The total Hamiltonian $\hat{H}(t)$ of a controlled system is (Giannelli et al., 2022b):

$$\hat{H}(t) = \hat{H}_0 + \sum_i a_i(t)\hat{H}_i, \qquad (10)$$

where $\hat{H}_0$ is the drift Hamiltonian, $a_i(t)$ are time-dependent control actions, and $\hat{H}_i$ are control Hamiltonians. In open quantum systems, environmental interactions lead to non-unitary evolution, also sometimes described as non-coherent. The master equation (Davies, 1974; Dirr et al., 2009) captures this evolution as:

$$\frac{\partial \rho(t)}{\partial t} = -\frac{i}{\hbar}[\hat{H}, \rho(t)] + \mathcal{L}(\rho(t)), \qquad (11)$$

where $[\hat{H}, \rho(t)]$ denotes matrix commutation, and $\mathcal{L}(\rho)$ describes non-unitary evolution (e.g. spontaneous emission, dephasing, cavity decay, etc.). Fidelity is a common measure of similarity between quantum states. For arbitrary density matrices $\rho$ and $\sigma$, the fidelity (Jozsa, 1994) reads:

$$\mathcal{F}(\rho, \sigma) = \left( \mathrm{Tr}\sqrt{\sqrt{\rho}\sigma\sqrt{\rho}} \right)^2 , \qquad (12)$$

where Tr is the trace. In this paper, we evaluate the fidelity between a target state $\rho_{\mathrm{des}}$ and the final evolved state $\rho(t_f)$ to assess the effectiveness of the applied controls $a_i(t)$.

## C. Electronic $\Lambda$ Systems

A very common system configuration in quantum information contains two ground-states $|g_1\rangle , |g_2\rangle$, coupled by a common excited state $|e_1\rangle$, as is required for the implementation of many quantum population transfer protocols, such as **St**imulated **R**aman **A**diabatic **P**assage (STIRAP) (Vitanov et al., 2017). We also include an additional excited state $|e_2\rangle$, detuned positively by an amount $\Delta_X$ from $|e_1\rangle$ to show the effect of crosstalk due to a coupling to an undesired transition. This configuration is ubiquitous and arises naturally in colour centres, quantum dots or other electronic quantum systems. An explicit energy level diagram is provided in Fig. 6. $\Omega_{P/S}$ denote the Rabi frequencies of the Pump and Stokes pulses respectively and $\Delta_{P/\delta}$ are the detuning of the Pump pulse from resonance as well as the two photon detuning respectively. The Hamiltonian $H_\Lambda$ used to model the unitary dynamics, defined in the basis $(|g_1\rangle , |g_2\rangle , |e_1\rangle , |e_2\rangle)$, after an application of the rotating wave approximation reads:

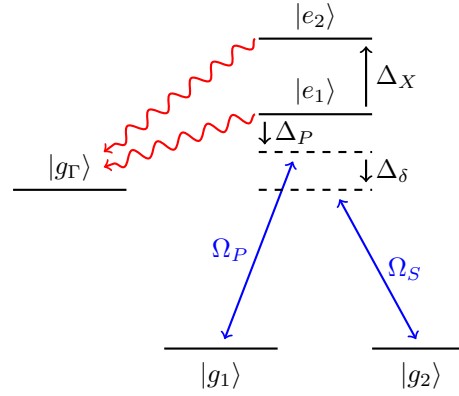

Figure 6: Energy level diagram for four level $\Lambda$ system with state $|e_2\rangle$, detuned positively by $\Delta_X$ from $|e_1\rangle$, to which cross talk is suppressed. There is an additional state $|g_\Gamma\rangle$ which does not partake in the unitary dynamics, but to which the excited states decay (cf. red dotted lines). This gives rise to a lower bound in attained population transfer fidelities. The laser couplings from Stokes and Pump laser are shown in blue.

$$H_\Lambda/\hbar = \begin{bmatrix} 0 & 0 & \frac{\Omega_P}{2} & \frac{\Omega_P}{2} \\ 0 & \Delta_P - \Delta_\delta & \frac{\Omega_S}{2} & -\frac{\Omega_S}{2} \\ \frac{\Omega_P}{2} & \frac{\Omega_S}{2} & \Delta_P & 0 \\ \frac{\Omega_P}{2} & -\frac{\Omega_S}{2} & 0 & \Delta_P + \Delta_X \end{bmatrix} \qquad (13)$$

All Rabi frequencies $\Omega_{P/S}$ are real. Additionally we include a sink state to which spontaneous emission occurs which couples equally to both excited state with rate $\Gamma/\sqrt{2}$, this is realistic insofar as spontaneous emission can always occur to states outside the manifold of interest, but as we do not consider spontaneous emission to $g_1$ or $g_2$ we obtain lower bounds on any population transfer fidelities $\mathcal{F}$. The Lindbladian operator reads; $\Gamma/\sqrt{2} |g_\Lambda\rangle \langle e_i|$.

The initial state is always fixed as $|g_1\rangle$, and the target (final) state is typically chosen as $|g_2\rangle$. However, the protocol can also be extended to other target states, such as the superposition state $|+\rangle = \frac{1}{\sqrt{2}}(|g_1\rangle + |g_2\rangle)$, as discussed in Ref. (Vitanov et al., 1999). More generally, the method allows for arbitrary superposition angles $\theta$, but in this work, we focus on the case $\theta = \pi$, which corresponds to full population transfer to $|g_2\rangle$. This scenario is widely studied and well-documented in the literature (Vitanov et al., 2017).

For the $\Lambda$ system we introduce an additional reward term which reads $-w_x \cdot (\langle e_1\rangle + \langle e_2\rangle)$. This assigns lower rewards to non-coherent dynamics, since we seek coherent population transfer and speeds up the learning dynamics.

We showcase two particular reference pulses for different pulse areas in Fig. 3. Trade-offs between pulse areas and

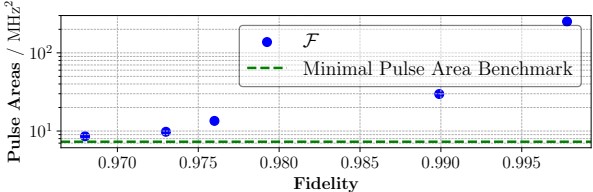

Figure 7: We sweep $w_A$, the pulse area penalty weight (cf. (3)) over a range of values $[0, 0.1, 0.25, 1, 2]$ and approach minimal pulse area with respect to Norambuena et al. (2023) (green dotted line) whilst achieving significantly higher fidelities $\mathcal{F} > 0.83$ (cf. Norambuena et al. (2023)). There is a clear trade off and lower pulse areas generally adversely affect fidelities.

population transfer fidelity are shown in Fig. 7. It is show that we approach the lower pulse area limit described in Ref. (Norambuena et al., 2023). We also show robustness to time-dependent noise in Fig. 8.

## D. Rydberg gates

We first consider a Rydberg gate based on a single laser excitation which is near resonant with the ground-state qubit $|1\rangle$ and Rydberg level $|r\rangle$ transitions. Following the implementation experimentally shown in (Levine et al., 2019) and the Hamiltonian definition given in (Pagano et al., 2022) the Hamiltonian for the one-photon Rydberg gate $H_{r_1} = H_0 + H_{\text{int}}$ reads:

$$
\frac{H_0}{\hbar} = \sum_{i=1}^{1} \left[ \frac{\Omega}{2} \big( |r\rangle \langle 1|_i + |1\rangle \langle r|_i \big) - \Delta |r\rangle \langle r|_i \right]
$$

$$
\frac{H_{\text{int}}}{\hbar} = B |r,r\rangle \langle r,r|
$$
(14)

Here $\Omega(t)$ and $\Delta(t)$ are real amplitudes and detunings of a Rydberg laser and $B$ describes the dipole blockade strength. The Linbladian terms are described by the addition of a sink state $g_\Gamma$ which imposes a lower bound on fidelity since any population which spontaneously decays leaves the computational subspace, as for the $\Lambda$ system. They read; $\sum_i \Gamma_r(|g_\Gamma\rangle \left( \langle r,i| + \langle i,r| \right) + \Gamma_r(|g_\Gamma\rangle \langle r,r|)$, where $\Gamma_r$ describes the decay rate of the Rydberg level. Many optimisation protocols consider $B \to \infty$, since the Rydberg gate operates in the regime $\Omega << B$ which precludes coupling of both qubits to $|r\rangle$, however we fix $B$ to a finite but realistic value in the range of hundreds of MHz (Pagano et al., 2022; Pelegrí et al., 2022; Sun, 2023).

One of the drawbacks of this implementation, as described in the main text however, is that it is not particularly robust in the face of signal imperfections and noise, such as atomic motion, laser intensity of frequency fluctuations. Some

work has been done to improve its robustness to quasi-static errors (Jandura et al., 2023; Mohan et al., 2023), but this often comes at the expense of the overall gate duration (Fromonteil et al., 2023) which is undesirable. Using the physics of a two photon process (similar to the $\Lambda$ system dynamics) we follow the Hamiltonian definition $H_{r_2} = H_{0,2} + H_{\text{int},2}$ for a two-photon Rydberg gate given in (Sun, 2023) (where h.c. denotes the hermitian conjugate):

$$
\frac{H_{0,2}}{\hbar} = \frac{\Omega_P(t)}{2} |10\rangle \langle e0| + \frac{\Omega_S(t)}{2} |e0\rangle |r0\rangle + \text{ h.c.}
$$
$$
+ \Delta_P(t) |e0\rangle \langle e0| + \Delta_S(t) |r0\rangle \langle r0|,
$$
(15)

with time-dependent Rabi frequencies $\Omega_P(t), \Omega_S(t)$, and values for the one photon detuning $\Delta_P$ and two-photon detuning $\Delta_S$. The Hamiltonian terms for $|01\rangle$ follow analogously from symmetry considerations by swapping the qubit states in $H_{0,2}$.

The interaction Hamiltonian $H_{int,2}$ for the state $|1,1\rangle$ consists of the atom light interaction as well as the dipole-dipole interaction akin to (14). A basis transformation simplifies the Hamiltonian, the new basis states read $|\tilde{e}\rangle = (|e1\rangle + |1e\rangle)/\sqrt{2}, |\tilde{r}\rangle = (|r1\rangle + |1r\rangle)/\sqrt{2}$ and $|\tilde{R}\rangle = (|re\rangle + |er\rangle)/\sqrt{2}$, after the rotating wave approximation, and effectively neglecting $|ee\rangle$, as we are in the regime where $\Delta_P >> \Delta_S$, $H_{int,2}/\hbar$ can be expressed as:

$$
\frac{H_{int,2}}{\hbar} = \frac{\Omega_P(t)}{\sqrt{2}} |11\rangle \langle \tilde{e}| + \frac{\Omega_S(t)}{2} |\tilde{e}\rangle \langle \tilde{r}| +
$$
$$
\frac{\Omega_P(t)}{2} |\tilde{r}\rangle \langle \tilde{R}| + \frac{\Omega_S(t)}{\sqrt{2}} |\tilde{R}\rangle \langle rr| + \text{ h.c. } +
$$
$$
\Delta_P(t) |\tilde{e}\rangle \langle \tilde{e}| + (2\Delta_P(t) + B) |rr\rangle \langle rr| +
$$
$$
(\Delta_P(t) + \Delta_S(t)) |\tilde{R}\rangle \langle \tilde{R}| + \Delta_S(t) |\tilde{r}\rangle \langle \tilde{r}|
$$
(16)

Parameters $\Omega_{S/P}, \Delta_{S/P}, B$ are defined as in (14). The Lindbladian decay terms for the two photon Rydberg gate are described similarly as for the one photon Rydberg gate. They read; $\sum_i \Gamma_r(|g_\Gamma\rangle)(\langle r,i| + \langle i,r|) + \Gamma_r(|g_\Gamma\rangle \langle r,r|) + \sum_i \Gamma_e(|g_\Gamma\rangle (\langle e,i| + \langle i,e|) + \Gamma_e(|g_\Gamma\rangle \langle e,e|)$, where $\Gamma_r$ describes the decay rate of the Rydberg level and $\Gamma_e$ the decay of the excited level $|e\rangle$ where for typical atoms $\Gamma_e >> \Gamma_r$.

Akin to the $\Lambda$ system we introduce an additional reward term which reads $-w_x \cdot (\langle rr\rangle + \langle \tilde{e}\tilde{e}\rangle + \langle \tilde{r}\tilde{r}\rangle + \langle rr\rangle + \langle \tilde{R}\tilde{R}\rangle)$. This assigns lower rewards to non-coherent dynamics, since we seek coherent population transfer and speeds up the learning dynamics.

The fidelity $\mathcal{F}_R$ is defined by the Bell state fidelity as is common in optimisation protocols of the Rydberg gate (Saffman et al., 2020; Jandura et al., 2023):

$$
\mathcal{F}_R = \frac{1}{16} |1 + \sum_{\psi_q^0 \in 10,01,11} e^{-i\theta_q} \langle q\rangle \psi_q^0|^2,
$$
(17)

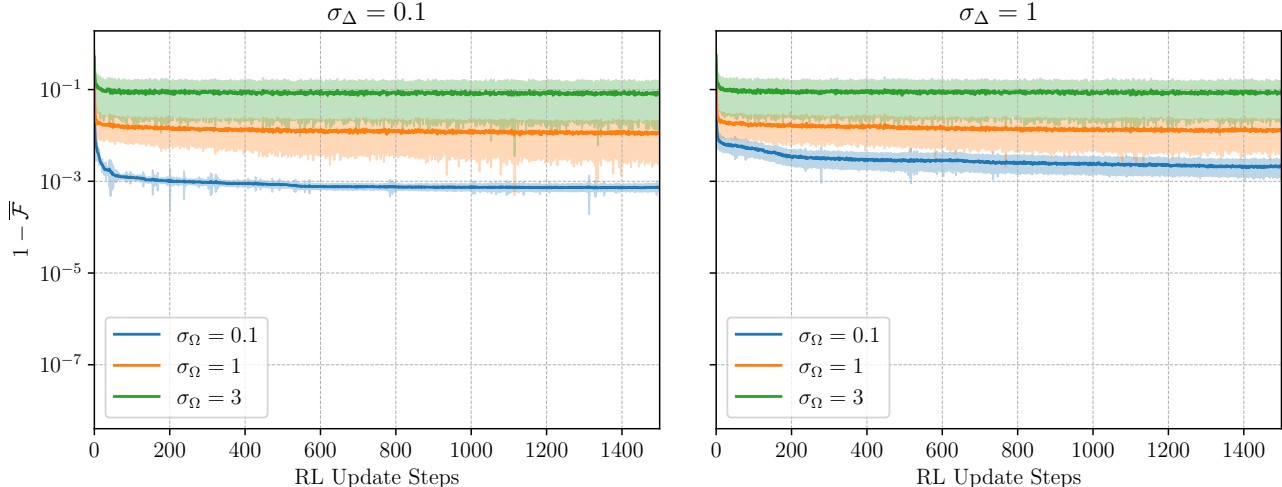

Figure 8: Robustness of PPO for $\Lambda$-system to randomly generated noise with $\mu_\Delta = 0.1$ MHz (left) and $\mu_\Delta = 1$ MHz (right) plotted on a logarithmic scale and averaged over multiple seeds. The solid lines show the average infidelity, while the shaded regions indicated one s.d. parallel environments. For small noise levels $F > 0.999$ as shown in the left plot for $\mu_\Delta = 0.1$ MHz, but as it increases fidelities drop to just below 0.97. Robustness to larger levels of noise is addressed with multi-step feedback in Sec. 5.4.

where $q$ is the simulated computational basis state. Without loss of generality, the optimisation here focusses on the case where $\theta_q = 0$, except $\theta_{1,1} = \pi$, this is particularly useful insofar as it does not require additional single qubit rotations (in comparison to a general $CZ$ gate) and does not introduce any further complexity associated with additional rotations. In practice, the computational complexity of simulating the Rydberg gate can be reduced by focusing on initialising the system in the coherent superposition state $|+,+\rangle$. The optimisation then targets both the populations and the relative phases of the resulting superposed states, ensuring they match the desired angles $\theta_i$ and maintain equal amplitudes. This approach effectively reproduces the transformation expected from an ideal logical C-Z gate:
$\frac{1}{\sqrt{4}}(|00\rangle + |10\rangle + |10\rangle - |11\rangle)$.

As described in the main text, we focus on the implementation of a two-photon Rydberg gate. For this, we fix the detuning of the pump pulse to a constant value, since a time-dependent frequency chirp offers no advantages in terms of achievable maximum fidelities, so we merely optimise its constant value. Over a duration of $0.5\mu s$ we fix $\Omega_S$ to a maximum value of $40$ and $\Omega_P^2/(2\Delta_P)$ (the effective Rabi frequency) to a maximum value of $56.6$ with a pump detuning of $2.5$ GHz and obtain an optimal control signal which is shown in Fig. 10. It shall be noted that the signals are different from results in the literature since we impose the realistic constraint of amplitudes to start and end at zero amplitude compared to (Sun, 2023). The optimal time-dependent control signals for a direct realisation of a C-Z gate are shown in Fig. 10. Simulated randomised benchmarking with 100

randomly generated initial states yields very similar fidelities to those obtained from the definition given in 17 with the $0.25\mu s$ C-Z gate yielding $\mathcal{F} \approx 0.99957 \pm 0.0003$ (versus $\approx 0.99958$ as obtained by taking the Bell state fidelity). Moreover, we observe a mean fidelity of $\overline{\mathcal{F}} > 0.999$ for time-dependent unbiased amplitude noise of $\sigma_\Omega = \pm 1$ MHz and frequency noise of $\sigma_\Delta = \pm 20$ MHz over 10 different randomly generated noise samples (cf. Eq. 25) for further details.

Following remarks made in Ref. (Sun, 2023) we reiterate that we show increased resilience to noise and achieve fidelities in excess of 0.999 even with significant levels of time-dependent noise, spontaneous emission (using realistic parameters for a $^{87}$Rb (Sun, 2023) atom) and a finite blockade strength of 500 MHz. Moreover, our control solution is robust in fidelity across a variety of smaller blockade strengths.

## E. Transmon Qubit Reset

Methods for unconditional Transmon qubit reset with fixed-frequency devices involve the coupling of a Transmon to a resonator through which excitations decay quickly. One particular hardware efficient protocol is based on a cavity-assisted Raman transition utilising the drive-induced coupling between $|f0\rangle$ and $|g1\rangle$, where $|sn\rangle$ denotes the tensor product of a Transmon in $|s\rangle$ and a readout resonator mode in the Fock state $|n\rangle$. By driving the Transmon simultaneously at the $|e0\rangle \leftrightarrow |f0\rangle$ transition and the $|f0\rangle \leftrightarrow |g1\rangle$ transition, we can form a $\Lambda$ system in the Jaynes-Cummings

ladder which can be used to reset the Transmon through fast single photon emission. The Transmon reset Hamiltonian is given by

$$
\frac{H}{\hbar} = \chi a^\dagger a q^\dagger q + \frac{g\alpha}{\sqrt{2}\delta(\delta + \alpha)}\Omega(t)(q^\dagger q^\dagger a + \text{h.c. }) + \\
(\Delta(t) + \delta_S(t))q^\dagger q \tag{18}
$$

where $a(a^\dagger)$ is the resonator lowering (raising) operator, $q(q^\dagger)$ the Transmon lowering (raising) operator, $\chi$ the Transmon-resonator dispersive shift, $\alpha$ the Transmon anharmonicity, $g$ the Transmon-resonator coupling rate, $\delta$ the difference in the Transmon and resonator resonant frequencies, $\Omega(t)$ the Transmon drive amplitude, $\Delta(t)$ the Transmon drive detuning, and $\delta_S(t)$ the drive-induced stark shift. As determined in Zeytinoğlu et al. (2015), this stark shift is to first order quadratic in the drive amplitude, $\delta_S(t) = k\Omega^2(t)$. For the Transmon mode we consider three levels $|g, e, f\rangle$ coupled with a two level resonator. We neglect self-Kerr terms in the resonator mode as we target single photon populations where such non-linearities are not significant.

The Lindbladian for the Transmon reset simulation is given by

$$
\dot{\rho} = -i\left[H_S, \rho\right] + \kappa\mathcal{D}[\rho] + \Gamma\mathcal{D}[\rho] \tag{19}
$$

with $\kappa$ describing the resonator decay rate, and $\Gamma$ the Transmon decay rate.

We construct the Transmon reset environment to match the physical parameters in Egger et al. (2018a), with maximum drive amplitudes of 330 MHz, however with an additional small detuning control of up to $\pm 100$ kHz for frequency corrections. To represent bandwidth constraints, we add a Gaussian convolution of duration 14ns to the amplitude and detuning defined in (20). We use the same reward function as in previous environments with a calibrated max-steps limit of 900, and we neglect the pulse area penalty.

We first optimise the reset for a higher qubit lifetime of $T_1 = 500$ us, representing the Transmon lifetimes currently attainable in experiment. Optimal waveforms and corresponding Transmon populations are shown in Fig. 4, where the RL Pulse can achieve fidelities of 0.9997 even with realistic bandwidth constraints. Notably, we find the RL agent consistently produces Gaussian-square like waveform for the drive amplitude, satisfying the high amplitude reset rate and optimising its smoothing. Novelty is observed in the time-dependent detuning, which first stays at a constant frequency throughout the drive until at reset a quick shift is observed from negative to positive. This results in the overall waveform correcting dynamic stark-shifts induced

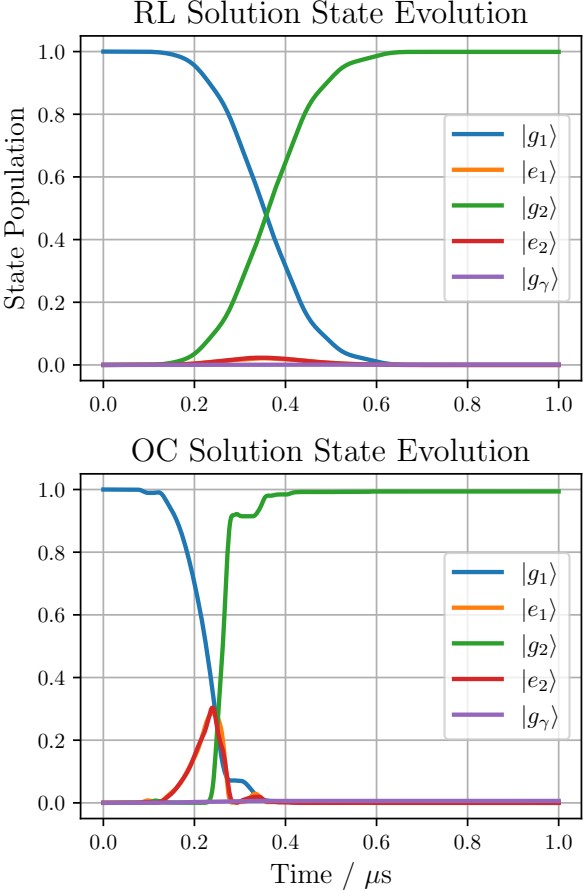

Figure 9: Comparison of quantum state dynamics for our RL solution (top) and Optimal Control solution (bottom). The RL solution exhibits a clearer interpretation of the optimal state dynamics with a smooth population of $|g_2\rangle$ and low excited state population $|e_i\rangle$. The optimal control solutions achieves high final population of the state $|g_2\rangle$, but the time evolution of the quantum states do not offer such a clear interpretation of the optimal time dynamics of the system.

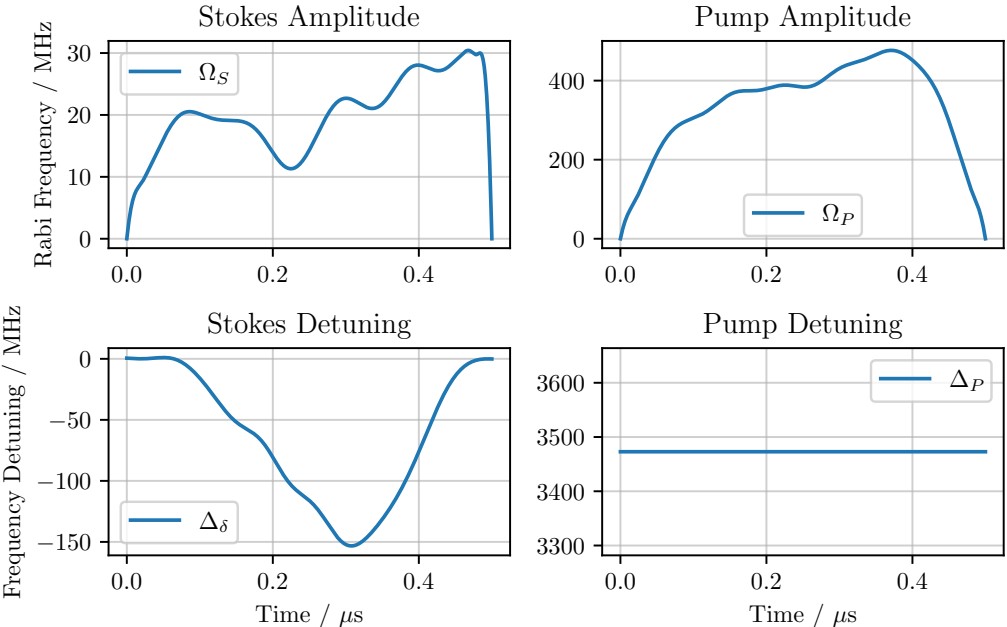

Figure 10: We show optimal signals for a two photon Rydberg gate directly realising a C-Z gate, with amplitudes (i.e. Rabi frequencies) for Stokes and Pump pulses in MHz shown in the top row. The effective maximum Rabi frequency of the pump pulse ($\Omega_P^2/2 * \Delta_P$) is $\approx 20$ to match that of the Stokes pulse. Detunings of the Stokes and Pump pulse are shown in the bottom row. Note the symmetry of the Stokes detuning in time which shows a semblance of a reflection symmetry about its centre which ensures that a relative $\pi$ phase is acquired between the basis states (cf. (17)) and their populations largely return to their initial values. This pulse yields a fidelity of $0.99917$ for a $0.5\mu s$ duration and can be shortened to $0.25\mu s$ with all signals re-scaled by 2 which yields a fidelity of $0.99958$ since we are mainly Rydberg level lifetime limited, with a finite blockade strength of 500MHz. This pulse is also shown to be robust across a variety of noise levels in amplitude and detuning.

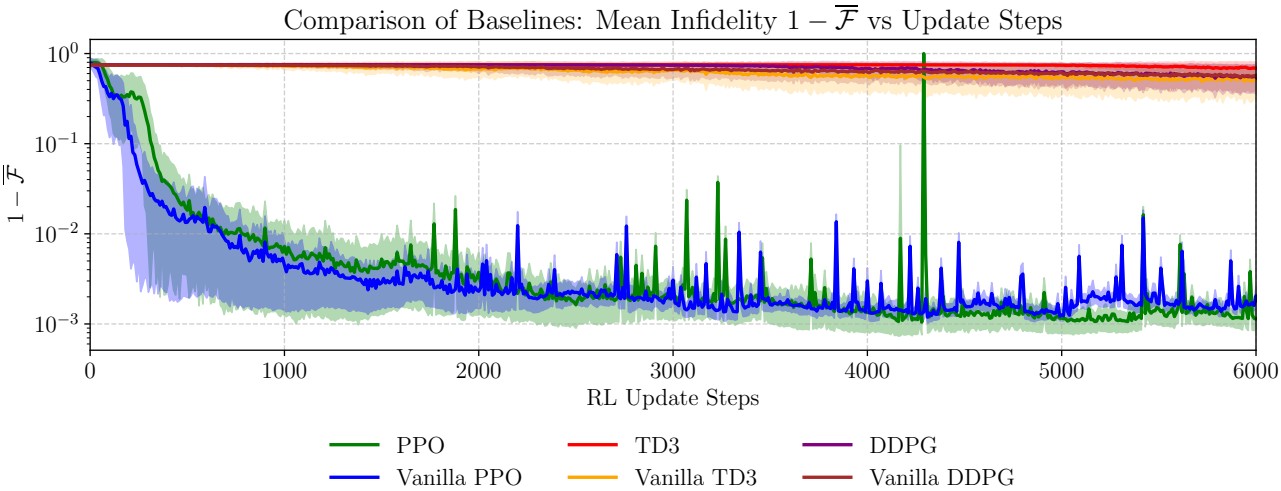

Figure 11: Comparisons of different RL Algorithms for Two Photon Rydberg Environment. As in Fig. 2 We compare three model-free RL algorithms — PPO (Schulman et al., 2017), DDPG (Lillicrap et al., 2016) and TD3 (Fujimoto et al., 2018). Each algorithm is run on the same Nvidia V100 GPU, using a single set of optimised hyper-parameters per method, with a common batch size (one s.d. shaded) for the same number of total steps. We compare standard (vanilla) versions of the algorithms with the reward function $L = \mathcal{F}$, to our constrained formulation, using the reward defined in Eq. 2 and incorporating step-size penalties (cf. Eq. 3). Vanilla PPO still performs reasonably, but it should be noted that it generates solutions which are extremely difficult to realise experimentally because of very high instantaneous bandwidth requirements, therefore, motivating a constrained formulation which also improves computational speed as shown in Fig. 1. PPO achieves the best overall performance, surpassing DDPG and TD3 by several orders of magnitude in mean infidelity. Furthermore, within a 24 hour GPU runtime period we could not achievable any reasonable fidelities with the alternative RL algorithms.

by the drive amplitude fall time, allowing for near ideal reset fidelities.

When reducing the Transmon lifetime to $T_1 = 48\mu s$ as used in prior experimental work, the RL agent produces a similar waveform that achieves 0.997 fidelity matching the ideal calibrated square evolution, and achieving higher results than a calibrated square pulse which gets 0.992 and the experimental results in Egger et al. (2018a) which achieved 0.983. The success in optimising over a range of Transmon $T_1$ lifetimes demonstrates that high fidelity unconditional reset can be achieved on current **N**oisy **I**ntermediate **S**cale **Q**uantum devices with advanced pulse control.

We further verify the RL solution quality in the context of a more significant Gaussian-smoothing kernel of 25ns and a qubit $T_1 = 500\mu s$, and find that it achieves high fidelities of 0.9995 while a standard square calibrated waveform deteriorates further to 0.9944 as errors arising from the uncorrected stark shifts become more significant.

### E.1. Heaviside Corrected Gaussian Square

For the $|f0\rangle \leftrightarrow |g1\rangle$ transition in the reset process, the RL agent consistently finds a Gaussian Square pulse for the drive amplitude which reminisces of prior works, however

with an additional Heaviside detuning profile as seen in Figure 4 which applies a frequency shift during the ring-down of the amplitude pulse.

This pulse, which we dub Heaviside-Corrected Gaussian Square (HCGS), directly corrects for a Hamiltonian which includes a drive-dependent stark-shift. Due to the finite ring-up time required for the amplitude, a negative frequency shift is applied to correct the positive amplitude-induced stark-shift. The negative frequency shift is applied throughout the reset until the ring-down. Before the ring-down of the square pulse, the Heaviside profile produces a positive detuning to correct for the negative amplitude-induced stark shift.

We note that this profile behaves quite similarly to past protocols such as DRAG where an additional phase component can be added to correct for unwanted Hamiltonian terms in the system. To further account for frequency bandwidth limitations, i.e. finite rise times for the phase control, the Gaussian Square duration $t_0$ and the Heaviside switch time $t_1$ can be at different points, with the Heaviside typically occurring a few nanoseconds earlier to account for the amplitude-driven stark shift.

Overall the HCGS reset pulse only requires 4 parameters, the amplitude $\Omega_0$ and duration $t_0$ of the Gaussian Square,

along with the detuning magnitude $\Delta_0$ and the Heaviside switch time $t_1$. The calibration procedure for Gaussian square pulse parameters has been detailed in previous works (Magnard et al., 2018; Egger et al., 2018a). Therefore, to calibrate the HCGS reset for real-world implementation of RL-optimised waveforms, only an additional sweep over detuning magnitude and switch time is needed.

## F. Experimental Feasibility

Although we focus on simulations of physical experiments we want to highlight below that it is physically feasible to implement our RL approach on real physical devices with fast gate times $< 1\mu$s in feasible time frames.

The amount of measurements is a function of the amount of RL steps required until convergence, specifically, for the Lambda system and Transmon system it is $N_{updates} * b$ where $b$ is the batchsize. For the noise free Lambda and Transmon system, this number is: $5100 * 256$ and $578 * 256$. For the Rydberg system, the physical number of measurements is $m * N_{updates} * b$ (where m = 4 - which is an overhead associated with the quantum state reconstruction), for the Rydberg environment this number is: $4 * 6800 * 256$.

In Baum et al. (2021) the authors show that they generate complete gate sets on a real device with $\mathcal{O}(10^6)$ measurements (i.e. individual steps) per gate, with gate times on the order of 100ns. The reported experimental runtime in (Baum et al., 2021) is on the order of hours, with API calls dominating this time overhead, showing the feasibility of scaling to large measurement numbers in real devices.

To reduce the number of experimental measurements, one can use a pre-trained policy from simulation which is then adapted to account for any possible mismatch between simulation and physical device parameters. In Ref. Li et al. (2025) the authors show that this reduces the number of on-device RL steps to a few thousand.

## G. Implementation Details

### G.1. Training Implementation Details

We leverage the Qiskit-Dynamics Solver interface (Puzzuoli et al., 2023) [2] for constructing both Hamiltonians and collapse operators, enabling the simulation of open quantum systems through the dissipative master equation. We employ the Diffrax ODE solver (Kidger, 2022) for quantum system simulation, which utilise adaptive step-sizing techniques to efficiently integrate the first-order linear differential equations, PureJAXRL for implementing PPO algorithms (Lu et al., 2022) and CleanRL (Huang et al., 2022) for TD3 and

---

[2]Similar functionality is available in Dynamiqs with comparable performance characteristics (Guilmin et al., 2024).

DDPG.

### G.2. Hyperparameters for RL Algorithm Benchmark

| Hyperparameter | PPO | TD3 | DDPG |
|---|---|---|---|
| ACTIVATION | relu6 | relu6 | relu6 |
| ANNEAL_LR | false | false | false |
| CLIP_EPS | 0.2 | – | – |
| ENT_COEF | 0 | – | – |
| GAE_LAMBDA | 0.95 | – | – |
| GAMMA | 0.99 | 0.99 | 0.99 |
| LAYER_SIZE | 256 | 256 | 256 |
| LR / LR_ACTOR | 0.0005 | 0.0003 | 0.0003 |
| LR_CRITIC | – | 0.0003 | 0.0003 |
| MAX_GRAD_NORM | 0.5 | – | – |
| MINIBATCH_SIZE | 32 | – | 32 |
| NUM_ENVS | 256 | 256 | 256 |
| NUM_MINIBATCHES | 8 | – | 8 |
| NUM_STEPS | 1 | 1 | 1 |
| NUM_UPDATES | 30,000 | 30,000 | 30,000 |
| UPDATE_EPOCHS | 4 | – | 4 |
| BATCH_SIZE | – | 256 | 256 |
| BUFFER_SIZE | – | 100,000 | 100,000 |
| EXPLORATION_NOISE | – | 0.15 | 0.15 |
| NOISE_CLIP | – | 0.5 | – |
| POLICY_FREQ | – | 1 | 1 |
| POLICY_NOISE | – | 0.2 | – |
| TAU | – | 0.01 | 0.01 |
| LEARNING_STARTS | – | 1,000 | 1000 |
| VF_COEF | – | – | 0.5 |

Table 2: Comparison of RL hyperparameters for comparison of PPO, TD3, and DDPG in Figs. 2 and 11.

We describe the RL hyper-parameters for our algorithm comparisons in Tab. 2. The vanilla algorithms use a loss function which only contains a linear fidelity term $\mathcal{F}$. The constrained formulations use a loss function as defined in Eq. 2 with appropriate smoothing hyper-parameters described in Fig. 13 and a max step size as inferred form Fig. 1.

### G.3. Benchmarking of Simulation Speed

Benchmarking absolute compute times across different hardware platforms, such as CPUs and GPUs, is challenging due to both systematic and random variations, even within the same architecture. Nevertheless, Fig. 12 highlights the advantages of GPU parallelisation for quantum simulations. We observe up to a two-order-of-magnitude improvement in speed per environment step, showcasing the significant performance benefits of running parallelised quantum simulations on GPUs, despite potential variability in the absolute timings.

Moreover, we compare the runtime of the different algo-

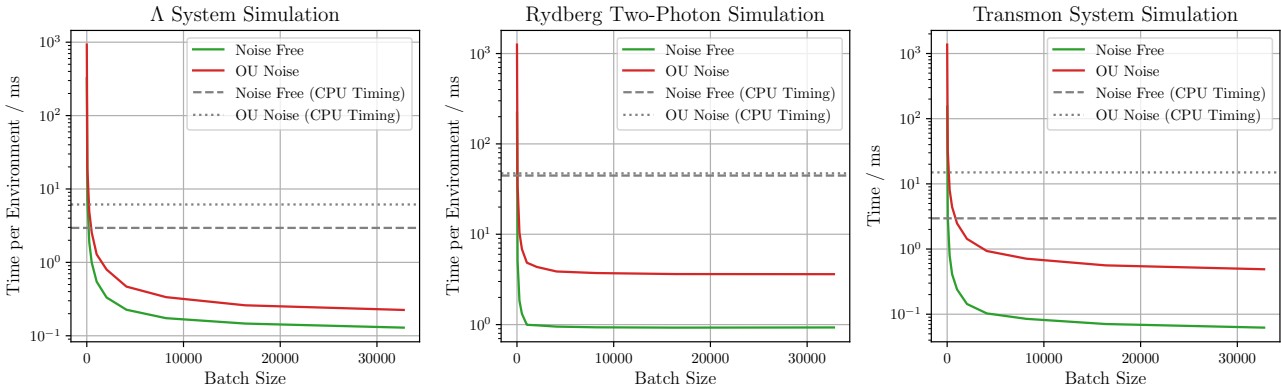

Figure 12: We compare the time per environment step for Qiskit Dynamics simulation across multiple environments ($\Lambda$ system, two photon Rydberg gate and Transmon) under noise-free and Orstein Uhlenbeck noise conditions. The left panel shows the $\Lambda$ system simulation timings, while the right panel illustrates the Rydberg two-photon simulation timings on a V-100 Nvidia GPU where we parallelise the simulation of several environments with different random actions and a fixed number of ODE solver steps= 4096. The solid lines represent the simulation times obtained with a GPU, while the dashed and dotted horizontal lines indicate the corresponding CPU timings (Apple Silicon M1) for Qiskit (noise-free and O-U noise, respectively). Simulation time per environment is plotted on a logarithmic scale and in the best case we get up to about two orders of magnitude improvement in simulation time per environment in a larger batch by moving to a GPU.

| Algorithm | Time to Conv. (s) |
|---|---|
| Our PPO | 239.1 |
| Reinforce RL (Brown et al., 2021) | 2284.8 |
| OC (BFGS) (Giannelli et al., 2022b) | 274.08 |
| Krotov (Goerz et al., 2019) | 3997 |

Table 3: Comparison of convergence times for different control algorithms which are benchmarked in Tab. 1 in the main text. Most of the algorithms we benchmarked did not run on GPU so we provide a CPU benchmark here (Mac M1 2020). We use a batch-size of 16 for Reinforce and our PPO implementation. We note that our implementation is largely faster due to the use of just in time compilation and automatic differentiation (Bradbury et al., 2018), but some speed up also comes from using a step size constraint on the ODE solver.

rithms benchmarked in Tab. 1 in the main text in Tab. 3 on a CPU and note that our implementation is faster. The RL algorithms implemented in Fig. 2 all ran on the same GPU and exhibit very similar compute times, hence we did not include an explicit timing comparison (the same holds for Fig. 11). However, we note that on the same GPU our constrained PPO formulation achieves $> 0.99$ fidelity in 39s compared to 4945s for DDPG and 4517s for TD3.

### G.4. Signal Processing & Analysis

The reinforcement learning (RL) agent samples actions from the interval $[-1, 1]$. For the Rabi frequencies $\Omega_{P/S}$, these actions are rescaled to the interval $[0, 1]$ to ensure all amplitudes remain positive and real. Phase variations are already accounted for through the optimisation of $\Delta_{P/\delta}$. No analogous rescaling is applied to the detunings $\Delta_i$.

Each action - whether an amplitude $\Omega_i$ or a detuning $\Delta_i$, denoted $a_i$ - is subsequently normalised by its respective maximum value, $\Omega_{\max}$ or $\Delta_{\max}$.

To encourage discovery of experimentally realistic pulse shapes, we apply additional smoothing and rescaling operations. The simulation timescale is fixed: $1\,\mu s$ for the $\Lambda$ system, $0.5\,\mu s$ for the Rydberg atom, and $0.2\,\mu s$ for the Transmon. All control signals are expressed in units of MHz. The parameters $\Delta_{P/\delta}$ and $\Omega_{P/S}$ are discretised into 50 time steps for the $\Lambda$ system and Rydberg atom, and 100 time steps for the Transmon. This discretisation provides a balance between control flexibility and computational efficiency. The actions $a_i$ are smoothed with a Gaussian convolution $(a * G)(t) = \int_{-\infty}^{\infty} \mathcal{A}(\tau) G(t - \tau)\, d\tau$, where the Gaussian function $G(t)$ is defined as:

$$G(t) = N(t_\sigma) \exp\left(-\frac{t^2}{2 \cdot t_\sigma^2}\right), \qquad (20)$$

where $t_\sigma$ defines the standard deviation and its value corresponds to the strength of the convolution filter. An ablation over this is provided in Fig. 13. This ensures that the

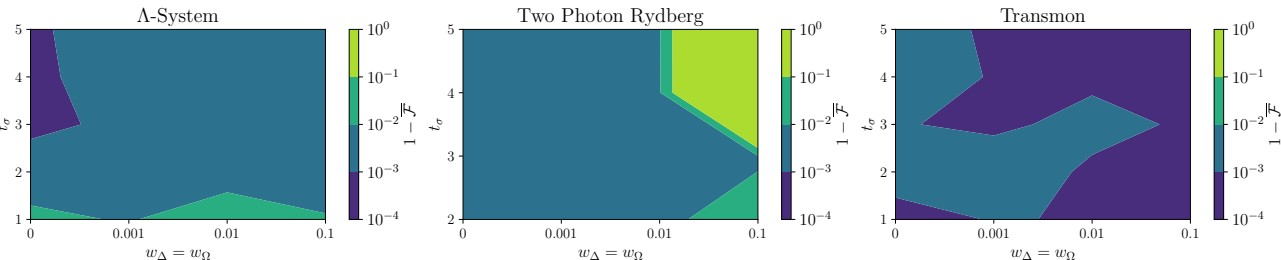

Figure 13: Ablation over smoothness penalty coefficients $w_\Delta = w_\Omega$ and filter standard deviation $t_\sigma$ for three different environments. Choice of smoothing parameters is important for learning policies with low mean infidelity $1 - \overline{\mathcal{F}}$ (averaged over 64 parallel environments). For some systems, like the $\Lambda$ system higher filter s.d. leads to lower infidelity, whilst for other like the Transmon higher smoothing penalties lead to lower infidelities. The relationship to experimental feasibility with limited bandwidth electronics is analysed in Fig. 14.

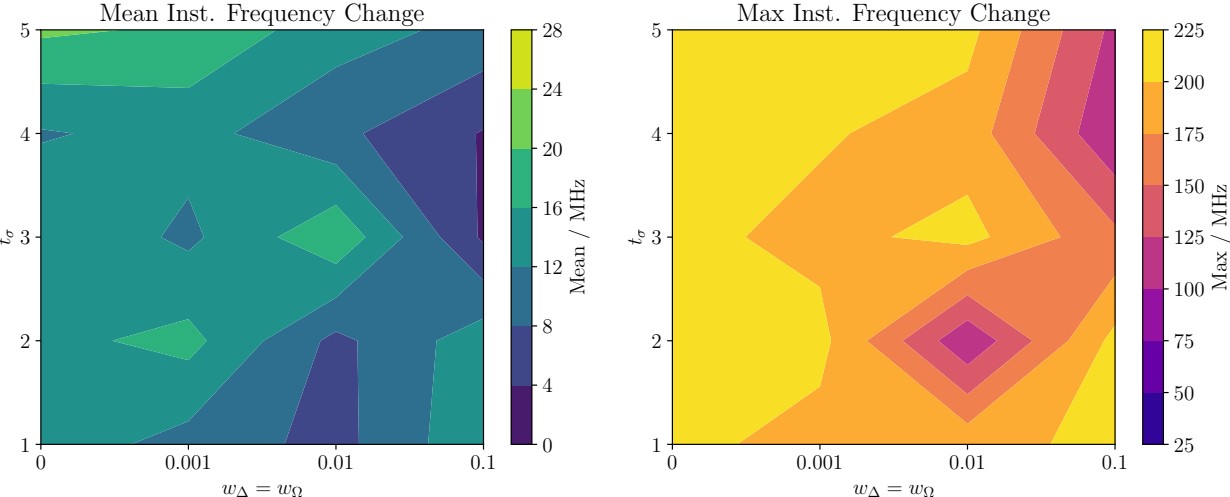

Figure 14: Using the optimised control signals for the $\Lambda$-system obtained from the smoothing hyper-parameter ablation study (Fig. 13), we analyse the instantaneous frequency change between successive output samples. This quantity is defined as $|\frac{dS}{d\text{sample}}|$, where $S = \Omega(t)\cos\left(2\pi\int_0^t \Delta(\tau)d\tau\right)$, and the signal $S(t)$ is digitally sampled at 1GSa/s, corresponding to state-of-the-art arbitrary waveform generator (AWG) capabilities (Keysight, 2024). We plot the mean (left) and maximum (right) instantaneous frequency changes across consecutive samples. Our results show that lower smoothing penalties and smaller kernel standard deviations lead to significantly larger maximum and mean instantaneous frequency variations. Instantaneous frequency changes of several hundred MHz pose serious challenges for experimental realisation, as changes exceeding 200MHz between samples are likely to induce severe signal distortions and fall outside the operational bandwidths of typical digital-to-analog converters (DACs), amplifiers, and other components in the signal chain. In contrast, signals with maximal instantaneous frequency changes of $\approx 100$MHz are far more compatible with standard experimental hardware. Thus, in addition to improving computational efficiency, the application of smoothing constraints is critical for ensuring that the optimised control signals are easily experimentally realisable. Similar results are found for the other physical systems but they are not further presented in favour of brevity.

generated time-dependent control signals are smooth and give rise to dynamics which can be solved in fixed number of time-steps, particularly at the beginning of the learning process when signals are randomly initialised. Pulse amplitude ends are always fixed at zero to ensure experimental viability with finite rise time effects, as signals cannot instantaneously start at non-zero amplitudes. Additionally, we use cubic spline interpolation (or linear interpolation for the Transmon) between action samples which is efficient for use with adaptive step size solver used for solving the master equation in different environments.

The pulse smoothness is defined in terms of different pulse smoothness functions. The first smoothing function is constructed by calculating the second derivative of $\mathcal{A}(t)$:

$$S_{der}(a(t)) = \int_0^{t_f} \left( \frac{d^2\mathcal{A}}{dt^2} \right)^2 dt. \tag{21}$$

An alternative smoothing function is defined in terms of the difference in output to that generated by a low pass Butterworth filter (Butterworth, 1930). This requires an expression of the filtered action which is the convolution of $\mathcal{A}(t)$ with the impulse response $h(t)$ of the Butterworth filter:

$$a_{\text{filter}}(t) = (h * A)(t) = \int_0^{t_f} h(t - \tau) A(\tau) \, d\tau \tag{22}$$

Calculating the difference with respect to the unfiltered signal, we get an expression for the low-pass smoothness with respect to a cut-off frequency $\omega_{max}$ and the filter order $n_{order}$:

$$S_{lp}(a(t), n_{order}, \omega_{max}) = \int_0^{t_f} |a_{\text{filter}}(t) - a(t)| \, dt. \tag{23}$$

It shall be noted that since all signals are discretised, the integrals decompose into discrete sums. The reference smoothness for an action is given by $S(B[t])$, where $B[t]$ is the Blackman window comprised of $n$ samples where $n$ also defines the number of signal samples corresponding to $\Omega_i$ or $\Delta_i$.

$$B[n] = \begin{cases} 0.42 - 0.5 \cos\left(\frac{2\pi n}{N-1}\right) + \\ 0.08 \cos\left(\frac{4\pi n}{N-1}\right), & 0 \leq n \leq N-1, \\ 0, & \text{otherwise.} \end{cases} \tag{24}$$

This choice is made as it is designed to have minimal spectral leakage, which means it suppresses high-frequency components effectively and mimics the smoothness of the signals that we are looking for. Penalising pulse smoothness is required because even after applying a convolution filter, we do not attain signals which exhibit low enough smoothness. The importance of generating *smooth* functions is three-fold. Firstly smoother waveforms are easier to experimentally implement with electronics with limited instantaneous bandwidth, as well as finite modulator rise times, and they are less vulnerable to signal chain delay or timing issues. This is shown in Fig. 14. Secondly, they are more interpretable in terms of the time evolution of the different quantum states. This is shown in Fig. 9. Thirdly, increased smoothness significantly speeds up the adaptive step size solver time which is particularly advantageous when working with limited computational resources or larger quantum systems.

Choosing the right smoothness penalty in the construction of the reward function is important as it can determine the learning speed and the extent to which realistic and interpretable controls are generated. We find, that a low-pass filter approach with the right cut-off frequency generally works well and provides the fastest learning of *smooth* signals as shown in Fig. 15. Other simpler smoothness functions such as the $L_1$ or $L_2$ norm are not considered because they were less well adapted for finding smooth signals that solved the quantum dynamics problems with a finite number of maximal adaptive solver steps.

Picking the right hyper-parameters for the Gaussian convolution filter standard deviation $t_\sigma$ defined in (20), as well as the right smoothing penalties $w_\Delta$ and $w_\Omega$ (cf. (3)) is crucial to ensure the optimal trade-off between smooth signal discovery to facilitate parallel optimisation, improved interpretability and discovery of high fidelity solutions. Overly strong signal smoothing or smoothing penalties result in the optimiser focussing largely on signal smoothness over fidelity of the quantum control task which is the primary objective. This is shown clearly in Fig. 13, where the $\Lambda$ system benefits from higher strict smoothing in form of a larger Gaussian kernel and higher weak smoothing in form of a larger pulse smoothness penalty, compared to the two photon Rydberg gate.

A final objective which competes with the fidelity, are the pulse areas $A(\Omega_i)$ and implicitly the pulse duration. $\Omega_{max}$ is limited physically by laser, RF or microwave power. Additionally, minimising pulse area is important for reducing the pulse energy and in turn the amount of heat introduced into the system, particularly for those quantum systems operating at cryogenic temperatures. Generally faster pulse sequences increase the clock cycles of a particular quantum operation which is desirable, but secondary to their fidelity, so implementing optimal control for some maximal amplitude $\Omega_{max}$ but with a minimal pulse area is considered in the example of a $\Lambda$ system. The baseline pulse area (cf. (3)), which is particularly relevant for the results shown in Fig. 7 and Fig. 3 is computed by comparing the generated pulse area $A = \int_{t=0}^{t=1} \Omega(t) dt$ to the area of a Blackman window

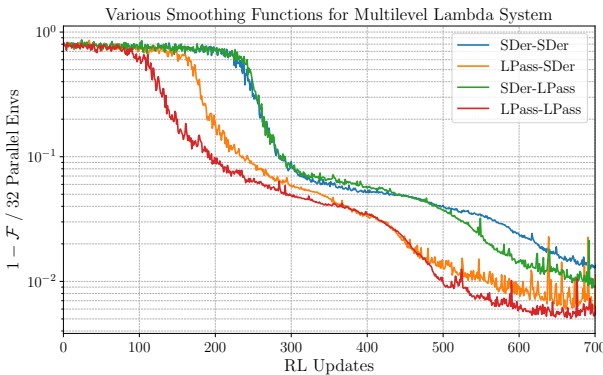

Figure 15: Comparison of different smoothing functions for mean infidelity (1- $\mathcal{F}$) across 32 parallel environments with different seeds plotted against the number of RL Updates for multi-level Lambda system. The legend corresponds to the type of smoothness penalty used where the ordering of the labels describes the amplitude and detuning smoothness functions respectively. One can observe that particularly for the amplitudes $\Omega_i$, using a low pass filter (LPass cf. (23)) instead of a second derivative penalty (SDer cf. (21)) allows for significantly sped up learning and also higher mean fidelity. For this ablation, the smoothness penalties $w_\Omega = w_\Delta$ are fixed to 0.001.

$A_B$ defined over the same timescale.

### G.5. Noise Model

We use an Ornstein-Uhlenbeck noise model defined with standard deviation $\sigma$ and mean $\mu$ which defines time-dependent noise in time $t$:

$$\nu_t = \nu_{t-1}(1 - \alpha^2) + \sqrt{2}\sigma X(t)\alpha + \sigma^2\mu, \qquad (25)$$

where $\alpha$ defines the characteristic time scale of the noise fluctuations and $X(t)$ is random Gaussian noise at time $t$ with a standard deviation of 1 and a mean of 0.

