# OpenReview forum: "Reinforcement Learning for Quantum Control under Physical Constraints"
_ICML.cc/2025/Conference — ICML 2025 poster_

### Official Review · Reviewer_K5by · 2025-03-12

**Overall Recommendation:** 4

**Summary:**

The authors address the problem of optimal quantum control using reinforcement learning (RL). Specifically, they define an RL framework that incorporates several real-world physical constraints to enhance performance.
First, they limit the agent’s possible actions to those that require a small number of simulation steps. This, they argue, has two key advantages:
(i) It improves training efficiency, as the fidelity component of the reward can be computed more efficiently.
(ii) It biases the agent toward more physically meaningful solutions due to a connection with adiabatic theory.
Second, they employ reward shaping to encourage the agent to generate smoother control signals, achieving similar benefits as (i) and (ii) above.
Finally, they benchmark their approach on several experimental settings and demonstrate strong performance.

**Claims And Evidence:**

Yes, the claims are clearly stated, and the provided evidence is sufficiently convincing.

**Essential References Not Discussed:**

Not that I am aware of.

**Experimental Designs Or Analyses:**

The experiments appear to be reasonable benchmarks for the problem. However, as I am not an expert in optimal quantum control, I cannot accurately assess the difficulty of these benchmarks or whether the achieved performance represents the state of the art.

**Methods And Evaluation Criteria:**

The experiments appear to be reasonable benchmarks for the problem. However, as I am not an expert in optimal quantum control, I cannot accurately assess the difficulty of these benchmarks or whether the achieved performance represents the state of the art. Nonetheless, they do beat rather involved schemes by a significant margin.

**Other Comments Or Suggestions:**

None aside from issues mentioned in Other strenghths and weaknesses.

**Other Strengths And Weaknesses:**

From a technical standpoint, the contributions are somewhat limited, as the approach builds on a standard RL framework for quantum optimal control, incorporating two novel techniques (limiting the agent’s action space and shaping the rewards). However, if these modifications lead to significantly better experimental results compared to the state of the art, this would still constitute a valuable contribution. Unfortunately, I am not able to assess how impressive the experimental results are due to my limited familiarity with the field.
A second concern is also honestly raised by the authors in the Limitations and Impact statement. Due to data (interaction) driven nature of the approach this method will require huge numbers of uses of a quantum device and/or unrealistically strong simulators.
It seems the only way this would really work would be in the far future, or given access to very good models of the system.
However the latter should be impossible by virtue of the hardness of the simulation of quantum systems.
So it is not entirely clear what the more immediate impact of this line of approaches will be.
There would clearly be value in pushing ML/RL machinery, but I don't think the strenghts of this contribution are there. We do learn interesting facts about the relevance of rescticting to some simpler-to-simulate actions in the search space, but it is not clear to me to what extent this necessarily implies we will be far from globally optimal solutions.

**Questions For Authors:**

None aside from issues mentioned in Other strenghths and weaknesses.

**Relation To Broader Scientific Literature:**

The questions are central to control, and extensively investigated; the work is well connected and the connections are listed.
The authors highlight the novelty of their work in two key aspects: (i) restricting the RL agent’s action space and (ii) defining a novel reward function. Through these mechanisms, they effectively incorporate physical constraints into the RL framework.

**Theoretical Claims:**

There are no critical theoretical claims in the paper that I was unable to verify.

---

> ### Author Rebuttal · Authors · 2025-04-01
>
> ## Reviewer K5by
>
> Thank you for recognising our incorporation of physical constraints, which significantly advance the state of the art, as a valuable contribution.
>
> ### Weaknesses
>
> > "contributions are somewhat limited, as the approach builds on a standard RL framework for quantum optimal control, incorporating two novel techniques (limiting the agent’s action space and shaping the rewards)."
>
> A4.1 While we agree that our work lies within applied machine learning, our implementation of physics-informed constraints enables a highly-parallelisable learning algorithm for arbitrary quantum systems, achieving state-of-the-art fidelities. This facilitates easy applications of RL to pulse-controlled quantum systems, aiding in the discovery of optimal and robust solutions, significantly surpassing other RL and non-RL methods.
>
> To highlight our contribution, we have conducted an ablation study in the noise-free Lambda system explained in Sec. 5.1. We replace PPO with two alternative algorithms, DDPG [1] and TD3 [2]. Additionally, we compare to a vanilla version of each algorithm, where the reward function includes only a linear fidelity term (as done in many previous works, e.g. [3]).
>
> The ablation results (see Table below) highlight the efficiency and effectiveness of our approach in achieving high-fidelity solutions. Notably, none of the vanilla algorithms achieve a fidelity above 0.99. Our PPO implementation significantly outperforms both DDPG and TD3, reaching fidelity >0.99 in less than 1% of the time. In our revised manuscript, we will include this baseline comparison along with a plot illustrating the learning dynamics.
>
> |Algorithm|Time to reach mean fidelity >0.99 (Nvidia P100-GPU)| Mean Fidelity (over 256 batches) after convergence|
> |-|-|-|
> |Constrained PPO (Ours)|39 s| 0.9997|
> |Vanilla PPO|never|0.989|
> |Constrained DDPG |4945 s|0.995
> |Vanilla DDPG |never|0.985|
> |Constrained TD3 |4517 s|0.992
> |Vanilla TD3 |never|0.625|
>
>
> > "Due to data (interaction) driven nature of the approach this method will require huge numbers of uses of a quantum device and/or unrealistically strong simulators."
>
> A4.2 While our method requires a large number of quantum device interactions, it has been demonstrated in [4] that this is feasible in real world systems, given their fast gate times.
>
> We also want to highlight that most quantum systems have extremely well understood theoretical models, once the model parameters are correctly identified, our method can be applied to real physical devices. This is demonstrated in prior work, where good agreement between experimental data and simulation results is shown [5, 6, 7].
>
> > I am not able to assess how impressive the experimental results are due to my limited familiarity with the field
>
> See A4.1 for RL baselines. We also wish to highlight that we outperform alternative optimisation methods on the respective problem settings in Sec. 5 with superior fidelities and noise robustness. Moreover, the methods we introduce for pulse-control of physical systems can find application in various fields of physical science such as in Optimising Chemical Reactions [9] and High Energy Physics [8].
>
> ### References:
>
> [1] Lillicrap et al. 2016, 'Continuous Control with Deep Reinforcement Learning', ICLR 2016
>
> [2] Fujimoto et al. 2018, 'Addressing Function Approximation Error in Actor-Critic Methods', ICML 2018
>
> [3] Bukov, Marin, et al. Physical Review X 8.3 (2018): 031086.
>
> [4] Baum, Yuval, et al. PRX Quantum, vol. 2, no. 4, 2021, p. 040324.
>
> [5] Magnard, P., et. al. Physical Review Letters, 121(6), 060502.
>
> [6] Willsch, Dennis. arXiv preprint arXiv:2008.13490 (2020).
>
> [7] Zhang, X. L. et.al. Physical Review A, 85(4), 042310.
>
> [8] Capuano F. et. al. arXiv preprint arXiv:2503.00499 (2025).
>
> [9] Zhou, Z. et. al. ACS Central Science (Vol. 3, Issue 12, pp. 1337–1344). American Chemical Society (ACS) (2017).

---

> > ### Comment · Reviewer_K5by · 2025-04-03
> >
> > I thank the authors for their comments and explanations.
> > The authors responded to three of my points: 1) unclear innovative step; 2) efficacy and; 3) importance of improvement.
> > Regarding 1) I appreciate the explication of the innovations, they are inline with what I originally understood.
> > Regarding 2) this helps but could the authors give explicitly the number of measurements needed for a more ambitious control problem?
> > Regarding 3) the level of improvement, the table helps, but is this a realistic representation of what one could expect in real devices? How many differing settings were the comparisons done in? Is "time to reach fidelity" a critical parameter?
> > Is getting an imporvement in the 3rd digit of precision a game changer? Could be?
> > I feel the points explained do improve my take on the paper, but not to the point I would increase the grade.

---

> > > ### Author Response · Authors · 2025-04-06
> > >
> > > > could the authors give explicitly the number of measurements needed for a more ambitious control problem?
> > >
> > > The amount of measurements is a function of the amount of RL steps required until convergence, specifically, for the Lambda system and Transmon system it is *num_rl_updates* * *batch_size*. For the noise free Lambda and Transmon system, this number is: 5100 x 256 and 578 x 256. For the Rydberg system, the physical number of measurements is *m* * *num_rl_updates* * *batch_size*  (where m = 4 -- which is an overhead associated with the quantum state reconstruction), for our Rydberg environment this number is: 4 x 6800 x 256.
> > >
> > > In a more complex setting, we expect this number to increase, for example in Ref. [1] the authors show that they generate complete gate sets on a real device with $\mathcal{O}(10^6)$ measurements (i.e. individual steps) per gate. The reported experimental runtime in [1] is in the order of hours, showing the feasibility of scaling to larger measurement numbers in real devices with fast gate times.
> > >
> > > > "Is this a realistic representation of what one could expect in real devices?"
> > >
> > > While experimental imperfections may slightly alter absolute fidelities, we expect to have a realistic representation of what one could expect in real devices and refer to papers that found close agreement between simulations and experimental results for the environment we consider, such as [2].
> > >
> > > > How many differing settings were the comparisons done in?
> > >
> > > We have run the full comparison with optimised hyperparameters for every algorithm in the Lambda system environment (Sec. 5.1) so far. We remark that the rebuttal time did not allow to run such extensive benchmarks in all environments. However, we observed similar trends between the different algorithms in the other environments and will add the full results to the final paper.
> > >
> > > > Is "time to reach fidelity" a critical parameter?
> > >
> > > Yes, because we show that our method allows training to 0.99 fidelity 100x faster than the other baselines. Effectively, this allows adjusting the control pulse to newly measured system parameters 100x faster than with the other baselines. This is significant as it minimises device time when doing real experimental control. It also is significant for more complex simulated environments with longer update steps, where - with limited compute time - baseline algorithms might never converge.
> > >
> > > > Is getting an imporvement in the 3rd digit of precision a game changer?
> > >
> > > Thank you for this great remark. We now clarify in the manuscript that an increase in fidelity from 0.99 to 0.999 is of great significance, as it can enable error-free quantum computing. We refer to seminal works in Quantum Computing [3,4,5], which find that for error-free quantum computations, with error correction, the physical error rate must be below a certain threshold value, often estimated to be around 10⁻³ or lower [3,4]. Thus, increasing fidelity from 0.99 to 0.999 surpasses this critical threshold, making error-corrected quantum computing feasible.
> > >
> > > [1] Baum et. al. PRX Quantum 2, 040324 (2021)
> > > [2] Xu, H. et. al. Nat Commun 7, 11018 (2016)
> > > [3] Fowler et. al. Phys. Rev. A 86, 032324 (2012)
> > > [4] Gottesman, D. (2002). An introduction to quantum error correction. In Proceedings of Symposia in Applied Mathematics (Vol. 58, pp. 221-236)
> > > [5] Preskill et. al. Quantum 2, 79 (2018)

---

### Official Review · Reviewer_8yQF · 2025-03-13

**Overall Recommendation:** 2

**Summary:**

This paper introduces a RL approach for quantum control under physical constraints, aiming to improve the fidelity and robustness of quantum control tasks in real-world scenarios.

Main Findings and Results:
1). The proposed physics-constrained RL algorithm achieves high-fidelity quantum control solutions across three different quantum systems. The fidelities exceed 0.999 across all tested systems, demonstrating superior performance compared to previous methods.
2). The method shows significant robustness to time-dependent perturbations and experimental imperfections.
3). By constraining the solution space to exclude control signals that induce overly fast quantum state dynamics, the algorithm improves computational scalability.

Main Algorithmic/Conceptual Ideas:
1). Physics-Constrained RL: The core idea is to incorporate physical constraints (e.g., maximum number of simulation steps)  directly into the RL framework.
2). Reward Shaping: The reward function is designed to incentivize high fidelity while penalizing solutions with large pulse areas and non-smooth control signals, which helps in discovering control policies that are easier to implement experimentally and less prone to errors due to signal imperfections.

**Claims And Evidence:**

The claims made in the submission are not well-supported by clear and convincing evidence. Specifically, the notion of "physics-constrained" is not sufficiently developed or explained. This lack of clarity undermines the overall persuasiveness of the paper.

(1) The term "physics-constrained" is used throughout the paper but is not adequately defined or justified. The authors mention incorporating physical constraints into the RL algorithm but fail to provide a detailed explanation of what these constraints entail and how they are specifically applied. Without a clear understanding of the constraints, it is difficult to assess their impact on the results and whether they truly enhance the robustness and efficiency of the quantum control solutions.

(2) While the authors present numerical simulations for three quantum systems, the results are not compared to a wide range of existing methods, especially the existing RL methods for quantum control.

(3) The computational efficiency claims are not convincingly demonstrated. The authors assert that their method improves computational efficiency by enabling parallel optimization, but they do not provide detailed comparisons with other methods in terms of computational resources and time. Without such comparisons, it is challenging to evaluate the actual benefits of their approach in terms of scalability and practical applicability.

**Essential References Not Discussed:**

There are several related works that are essential for understanding the context and significance of the key contributions, but are not currently discussed in the paper. For example, Universal quantum control through deep reinforcement learning (npj Quantum
Inf. 2019), Curriculum-based deep reinforcement learning for quantum control (IEEE TNNLS 2023).

**Experimental Designs Or Analyses:**

Not exactly. The experimental design and analysis in the paper are based on numerical simulations, which are appropriate for the initial validation of the proposed method.

**Methods And Evaluation Criteria:**

The proposed methods and evaluation criteria need more clarity, justification, and comprehensive validation to be considered appropriate for quantum control tasks. For example, the "physics-constrained" approach is vaguely defined, making it not clear to see how it specifically addresses quantum control challenges. The reward function's penalties lack clear justification, questioning their effectiveness in guiding the RL agent. Computational efficiency claims are unconvincing due to the lack of detailed benchmarking against other existing methods.

**Other Comments Or Suggestions:**

No.

**Other Strengths And Weaknesses:**

Strengths：
（1）The integration of physical constraints into the RL framework is a novel approach that enhances the practicality and robustness of quantum control solutions.
（2）It is a significant advancement of achieving high fidelity and robustness to noise.

Weaknesses:
(1) The paper relies solely on simulations without experimental validation. How to assess its practical applicability?
(2) More comprehensive benchmarking against existing methods especially the existing RL methods for quantum control would strengthen the paper's claims and demonstrate its advantages.

**Questions For Authors:**

1. How does the proposed method scale to larger quantum systems with more qubits or higher-dimensional state spaces?
2. How does the proposed RL approach compare with other existing RL methods for quantum control tasks?
3. How to verify the performance of the proposed approach on real physical systems?
4. How well does the proposed RL method generalize to other types of quantum systems?
5. How sensitive is the method to the choice of physical constraints such as maximum solver steps? How to choose proper physical constraints?

**Relation To Broader Scientific Literature:**

The paper's key contributions are well-aligned with the broader scientific literature on quantum control and RL.

**Theoretical Claims:**

Not applicable. The claims made are largely based on the results of simulations and the design of the algorithm, rather than on formal mathematical proofs.

---

> ### Author Rebuttal · Authors · 2025-04-01
>
> ## Reviewer 8yQf
>
> We thank the reviewer for recognising our integration of physical constraints into RL, which achieves high-fidelity, noise-resilient solutions.
>
> ### Weaknesses
> > "The term "physics-constrained" is used throughout the paper but is not adequately defined or justified."
>
> A3.1: Thank you for raising this point, we refer to A1.3 in the response to sD3o and provide additional detail in A3.7.
>
> > "The computational efficiency claims are not convincingly demonstrated. "
>
> A3.2: We acknowledge the lack of timing comparison in our work and provide it here for the results in Table 2 (Sec. 5.1) (with one additional comparison) for the noise-free λ System Optimisation. On a Mac M1 2020 CPU, our highly parallelisable implementation achieves up to 30x faster runtimes for a single run, while also yielding the highest fidelity.
>
> We further show speedups over the baseline time by up to two orders of magnitudes with GPU based parallelisation in Appendix Fig. 11.
>
> |Algorithm|Time to convergence| Mean Fidelity|
> |-|-|-|
> |Our RL Algo|239.1 s| 0.999|
> |Reinforce RL Algo [T.1]| 2284.8 s|0.93|
> |OC (BFGS) [T.2]|274.08 s|0.89
> |OC Krotov [T.3]|7216 s|0.99|
>
>
> ### Questions
>
> >How does the proposed method scale to larger quantum systems with more qubits or higher-dimensional state spaces?
>
> A3.3 Thank you for bringing this up, see response A1.4 to sD3o.
>
> >How does the proposed RL approach compare with other existing RL methods for quantum control tasks? [...] "the results are not compared to a wide range of existing methods, especially the existing RL methods for quantum control." [...] "For example, [2], [3]"
>
> A3.4: [2] uses discrete actions, unsuitable for our real-world quantum control problems requiring continuous actions. Hence we cannot benchmark against it, but have added it to our related work section.
>
> Unfortunately, [3] does not provide an open source implementation and the paper does not provide sufficient detail to allow reimplementation, hence, we are unable to benchmark against it.
>
> We want to highlight that we already included [3] in our related work section. We now include a more comprehensive discussion, highlighting its computationally less efficient approach of learning control signal parameters individually at each signal time step.
>
> We are confident that our work includes empirical comparisons to all relevant baselines, and we kindly ask the reviewer to specify if any comparisons were missing. We now also include benchmarks against DDPG and TD3 and refer the reviewer to A4.1 in response to K5by.
>
> >How to verify the performance of the proposed approach on real physical systems?
>
> A3.5 Most quantum systems have well established theoretical models. Given the system parameters, learned pulses can be directly applied to real devices; many prior works show good agreement between simulation and experiment [4,5,6].
>
> In future work, we are planning to extend our experiments to real world devices.
> Unfortunately, IBM has discontinued pulse-level access, so we are pursuing Rigetti [7] as an alternative.
>
> >How well does the proposed RL method generalise to other types of quantum systems?
>
> A3.6 Our model-free approach generalises to any quantum system that can be simulated. Our method can also be extended to black box sampling from real physical devices; the feasibility of this has been shown in [1].
>
> >How sensitive is the method to the choice of physical constraints such as maximum solver steps? How to choose proper physical constraints?
>
> A3.7 Our ablation study (App. Fig. 12 and 13) shows that larger smoothing kernel standard deviations and smoothing penalties boost fidelity while keeping signals within the bandwidth limits (typically a few hundred MHz) of standard electronics. In the revised manuscript, we will expand on App. Sec. F.3 to explain how we choose physical smoothing constraints that are consistent with electronic bandwidth limitations.
>
> Limiting solver steps improves compute time without affecting optimal dynamics if steps are sufficient. We set N_max based on the adiabatic condition, determined by the maximal effective Rabi frequency (N_max ≳ 1/Ω_eff) and increase it until infidelity significantly drops, as outlined in lines 206–214.
>
> ### Concluding remarks
> We kindly ask the reviewer if we’ve addressed their concerns by (1) adding RL benchmarks and distinguishing our work from existing RL work, (2) including the requested timing comparison, and (3) explaining physics-driven constraints and their effects. With these updates and new results, we politely ask the reviewer to reassess their score or highlight remaining issues.
>
> ### References:
> [1] Baum, Yuval, et al. PRX Quantum, vol. 2, no. 4, 2021, p. 040324.
>
> [2] Niu et al 2019, npj Quantum Information
>
> [3] Ma et al. 2023, IEEE TNNLs
>
> [4] Magnard, P., et. al. Physical Review Letters, 121(6), 060502.
>
> [5] Willsch, Dennis. arXiv:2008.13490 (2020).
>
> [6] Zhang, X. L. et. al. Physical Review A, 85(4), 042310.
>
> [7] Rigetti Computing. (2025). pyQuil 4.16.1.

---

### Official Review · Reviewer_Vwkd · 2025-03-13

**Overall Recommendation:** 3

**Summary:**

This paper explores the application of reinforcement learning (RL) for quantum control, introducing constraints aimed at improving learning efficiency. The authors present a rigorous approach to adapting RL for quantum applications and provide detailed reasoning behind the necessary modifications. The study focuses on a specific use case within quantum control, integrating a tailored set of constraints and optimizations to enhance RL performance.

### Update after rebuttal

The authors have provided a detailed and constructive response that addresses all previously raised concerns. They clarified the distinction between their pulse-level quantum control setting and prior circuit-level RL work, appropriately situating their contributions within the literature. Code has been made anonymously available and demonstrates compatibility with standard RL libraries. Additionally, they conducted new experiments benchmarking PPO against DDPG and TD3, showing clear advantages in both performance and sample efficiency. The authors also elaborated on their notion of “interpretable quantum state dynamics,” adding further clarity. In light of these clarifications and additions, I updated the score accordingly.

**Claims And Evidence:**

The authors make several claims regarding the effectiveness of their constrained RL approach for quantum control. While the paper provides strong theoretical and empirical reasoning for these adaptations, there are notable gaps in supporting evidence:
- The paper does not sufficiently discuss prior work on RL applications in quantum computing.
- Details on the specific RL algorithm employed and justification for its suitability to this problem are lacking.
- The absence of released code raises concerns about reproducibility and the ability to benchmark against alternative methods.

**Essential References Not Discussed:**

e.g., the following work on RL for Quantum Computing could have been considered:
- van der Linde et al., 2023: RL-based quantum compilation benchmarking
- Altmann et al., 2024: Challenges of RL in quantum circuit design
- Kölle et al., 2024: RL environment for quantum circuit synthesis
- Rietsch et al., 2024: RL for unitary synthesis of Clifford+T circuits

**Experimental Designs Or Analyses:**

The experimental setup is well-explained for its intended quantum control application. However:
- It is unclear whether the implementation supports standard RL frameworks, which would facilitate broader testing.
- The paper would benefit from additional benchmarking against existing RL-based quantum approaches to contextualize performance improvements.

**Methods And Evaluation Criteria:**

The proposed methods appear well-suited for the specific quantum control application, with clear justifications for introduced constraints. However, broader ML contributions are limited, as the study primarily focuses on improving RL’s applicability to quantum systems rather than advancing RL methodologies themselves. The evaluation could be strengthened by benchmarking against standard RL methods for quantum applications, incorporating comparisons to prior approaches referenced in the literature.

**Other Comments Or Suggestions:**

If targeting ICML, consider emphasizing broader ML contributions beyond quantum-specific constraints.

**Other Strengths And Weaknesses:**

Strengths:
- A rigorous and well-explained adaptation of RL for quantum control.
- Provides strong reasoning behind constraints and modifications.
- Addresses a specific and well-motivated quantum control problem.

Weaknesses:
- The paper is heavily skewed towards quantum-specific adaptations, limiting its ML relevance for ICML.
- Missing discussion of prior RL-based quantum computing research.
- No code provided, reducing reproducibility and practical applicability.

**Questions For Authors:**

What do the authors mean by "interpretable quantum state dynamics" in this context?

Do the authors plan to release their implementation, and if so, will it support standard RL libraries for benchmarking?

**Relation To Broader Scientific Literature:**

While the paper provides a rigorous adaptation of RL for quantum control, it does not sufficiently situate its contributions within the broader field of RL for quantum applications. Prior work such as [1-4] should be discussed to highlight differences and advancements beyond existing methods.

**Theoretical Claims:**

The paper does not primarily focus on new theoretical advancements in RL or quantum computing. However, the mathematical formulations appear sound.

---

> ### Author Rebuttal · Authors · 2025-04-01
>
> ## Reviewer vWkD
>
> Thank you for recognising our rigorous adaptation of RL for quantum control and well reasoned introduction of constraints.
>
> ### Weaknesses
>
> > "Missing discussion of prior RL-based quantum computing research". [...] "... Prior work such as [1-4] should be discussed..."
>
> A2.1: We thank the reviewer for highlighting these works; we will discuss them in the related work of the updated manuscript. We remark that these works do not address the same problem setting but instead focus on circuit-level quantum *compilation* (in a space of logical quantum gates) which is fundamentally different from the pulse-level *control* problem (in a space of physical control signals) that our work addresses. Hence, these works do not constitute approaches that we could benchmark against. We mention in the conclusion that expanding our pulse level control to multiple sequential pulses (i.e. a circuit) would be an interesting extension.
>
> > "No code provided, reducing reproducibility and practical applicability." [...] "The absence of released code raises concerns about reproducibility and the ability to benchmark against alternative methods.""
>
> A2.2: We acknowledge the reviewers' concern and are committed to reproducibility. To allow for a better review process, we have anonymously published the code under https://anonymous.4open.science/r/RL4qcWpc/README.md. We will release all code publicly upon acceptance, as stated in Section 5.0.1.
>
> > "The paper would benefit from additional benchmarking against existing RL-based quantum approaches..."
>
> A2.3: As pointed out in A2.1 above, the highlighted related works [1-4] do not address the same problem setting as our work.
>
> We are confident that our work includes empirical comparisons to all relevant baselines and politely ask the reviewer to specify if there were any missing comparisons.
>
> > "It is unclear whether the implementation supports standard RL frameworks, which would facilitate broader testing."
>
> A2.4: Our implementation supports any RL framework, as can be seen in the published code. To emphasise this, and to further motivate the choice of PPO, we have now conducted an ablation study on replacing PPO by DDPG [5] or TD3 [6]. Our implementation of PPO achieves fidelities of over 0.999 for the experimental setup described in Sec. 5.1, whereas both DDPG and TD3 achieve maximum fidelities of 0.995 and 0.994 and take 100x longer to reach >0.99 fidelity on the same GPU. See A4.1 in response to K5by for more details.
>
> > "Details on the specific RL algorithm employed and justification for its suitability to this problem are lacking."
>
> A2.5: Thank you for this feedback, we have added motivation for using PPO, an on-policy algorithm chosen for stability (less hyperparameter sensitivity) over DDPG [5] or TD3 [6]. Furthermore, we show that in our quantum control setting, PPO significantly outperforms alternatives, see A2.4 and A4.1 for more details.
>
> > "The paper is heavily skewed towards quantum-specific adaptations, limiting its ML relevance for ICML".
>
> A2.6: We believe that the paper's focus on physics-constrained RL for Quantum Control exactly fits the scope of ICML defined in the call for papers. Specifically, ICML calls for “Application-Driven Machine Learning [...] driven by needs of end-users in applications [...] such as physical sciences.” Also, see A4.1 for an ML specific benchmark.
>
> ### Questions
> > "What do the authors mean by "interpretable quantum state dynamics" in this context?"
>
> A2.7: It refers to the clear, interpretable time evolution of a quantum system under optimised control (e.g. minimal excited state population in a Lambda system), unlike complex, fast dynamics induced by many unphysical optimal control solutions. A plot contrasting these will be added to the revised manuscript.
>
> > "Do the authors plan to release their implementation... will it support standard RL libraries for benchmarking?"
>
> A2.8: As noted in A2.2, the code is now anonymously available and supports standard RL libraries, shown via ablations with DDPG [5] and TD3 [6] vs. PPO (see A2.4).
>
> ### Concluding remarks
> We kindly ask the reviewer if we were able to address their concerns by (1) clarifying the difference to [1-4], (2) making our code available, (3) demonstrating adaptability through comparisons with other RL algorithms. In light of these clarifications and additional results, we politely ask the reviewer to consider updating their score, or to point out further concerns.
>
> ### References:
>
> [1] van der Linde et al., 2023: RL-based quantum compilation benchmarking
>
> [2] Altmann et al., 2024: Challenges of RL in quantum circuit design
>
> [3] Kölle et al., 2024: RL environment for quantum circuit synthesis
>
> [4] Rietsch et al., 2024: RL for unitary synthesis of Clifford+T circuits
>
> [5] Lillicrap et al. 2016, 'Continuous Control with Deep Reinforcement Learning', ICLR 2016
>
> [6] Fujimoto et al. 2018, 'Addressing Function Approximation Error in Actor-Critic Methods', ICML 2018

---

### Official Review · Reviewer_sD3o · 2025-03-14

**Overall Recommendation:** 4

**Summary:**

The paper proposes a physics-constrained reinforcement learning algorithm to explore physically realizable pulses for quantum control tasks. The constraint on the pulses ensures smooth transitions and low energies, which result in noise robust pulses that may achieve higher fidelities.

Comprehensive experiments are conducted on three quantum control tasks and three quantum architectures, demonstrating the effectiveness of the proposed method.

**Claims And Evidence:**

Yes.

**Essential References Not Discussed:**

No.

**Experimental Designs Or Analyses:**

Yes.

**Methods And Evaluation Criteria:**

Yes.

**Other Comments Or Suggestions:**

Minor comments:
The second and third term in (2) has the same structure, which I believe is a typo.

**Other Strengths And Weaknesses:**

Pros:

1.	The framework considers a hard balance in quantum control theory: how to find robust pulses that drive the quantum system to a desired state with high precision. This is tackled by an easy but clever combination of constraint design and RL methods to achieve high-robustness and high-fidelity at the same time.

2.	The experiments include many interesting cases with multiple architectures, demonstrating the general feasibility of the proposed method. The resulting pulses look impressive in the sense of smoothness and pulse duration, both are essential in the real experiments.

3.	The training environment includes a simulation of the Lindbladian of the quantum system, which is precise but demanding computation. The framework makes optimization on the engineering side to maximize the parallelism to speed up the training procedure.

Cons:

1.	The proposed method requires precise calibration of the target system, and after each calibration the RL algorithm needs to be rerun to obtain high-quality pulses. I wonder, when integrated in a realistic system where calibrations are conducted routinely, how will the proposed RL algorithm perform (in the sense of run time and stability) in real devices.

2.	The experiments in the paper are conducted with numerical simulations. For a technique targeting quantum control, it is essential to validate the method on real quantum devices. I understand that it might be hard for the authors to access the limited hardware resource, while I still want to know if the proposed method can be applied to accessible commercial quantum devices, i.e., IBM’s devices.

3.	Following point 2, the constraint design in the framework is not convincingly justified in real device experiments. In this sense, the paper lacks an important discussion on the criteria for selecting these constraints. Though, I do not doubt that the included constraints represent important features of robust pulses.

4.	Since the RL algorithm relies on a full simulation of the system, it is hard in its current form to scale up, i.e. to more than 10 qubits.



The above weaknesses are a bit nitpicking and may be out of scope of the current paper. Overall, I enjoyed reading the paper and like its idea. It is well-executed research in the direction of AI for quantum science.

**Questions For Authors:**

See weaknesses.

**Relation To Broader Scientific Literature:**

The paper proposes a novel RL framework to generate more robust quantum control pulses, which is a hard task for quantum control and little discussed in the literature of quantum + AI.

**Theoretical Claims:**

There is no theoretical claim in the paper.

---

> ### Author Rebuttal · Authors · 2025-04-01
>
> ## Reviewer sD3o
> We appreciate your positive feedback on our work and are glad that our research in AI for quantum science was well received.
>
> ### Weaknesses
> > "The proposed method requires precise calibration of the target system, and after each calibration the RL algorithm needs to be rerun to obtain high-quality pulses. I wonder, when integrated in a realistic system where calibrations are conducted routinely, how will the proposed RL algorithm perform (in the sense of run time and stability) in real devices."
>
> A1.1: We thank the reviewer for this insightful question regarding the handling of system parameter changes after re-calibration. For small drifts the inherent robustness of our learned policy (achieved via training in noisy environments) can avoid requiring retraining. For large deviations in system parameters, our method can be adapted by training an RL policy that conditions on the current system parameters. This policy is trained by sampling environments with varying system parameters, enabling it to adapt to novel system parameters when deployed.
>
> We would like to point out an additional scenario that we observed in our experiments, where the learned policy can be attributed to symbolic equations (Appendix Section E.1), as is the case for the learned transmon reset pulse.
> Only requiring four parameters to define, the learned pulse can be easily tuned in a physical experiment and does not require re-training.
>
> > "The experiments in the paper are conducted with numerical simulations. For a technique targeting quantum control, it is essential to validate the method on real quantum devices... hard for the authors to access the limited hardware resource, while I still want to know if the proposed method can be applied to accessible commercial quantum devices, i.e., IBM’s devices."
>
> A1.2: Unfortunately, IBM does no longer allow pulse-level access to their devices. For future work, we are investigating alternative hardware providers which offer pulse level device access, including Rigetti [1] and IQM [2] to verify our solutions. Furthermore, we want to highlight that prior work found good agreement between numerical simulations of and experiments with transmon qubits [3,4].
>
> > "Following point 2, the constraint design in the framework is not convincingly justified in real device experiments. In this sense, the paper lacks an important discussion on the criteria for selecting these constraints..."
>
> A1.3: Thank you for this remark. We refer to the introduced method as "physics-constrained", as all constraints are derived from real, physical limitations.
>
> First, we constrain signal bandwidth (smoothness), and signal area, reflecting the limited instantaneous bandwidth of electronics and signal components in experiments, and limitations in available signal power and duration. We implement these constraints via the reward function (Eq.2), similar to the Lagrange Multiplier technique introduced in [5].
>
> Second, we constrain the policy to solutions that can be simulated within a predefined number of maximum solver steps. This constraint incorporates priors about the physical solution time scales into the algorithm (see lines 209-13), which incentivises adiabatic quantum state dynamics, yielding robust and interpretable solutions. Additionally, this constraint incentivises smoothness (requiring lower bandwidth) of signals, as smooth signals generally require fewer solver steps. Furthermore, the constraint on maximum solver steps significantly reduces computational demand.
>
> We will include this explanation in Sec. 4.1 in our revised manuscript to make it clearer for the reader. We will also add a figure in Appendix Sec. F.3 which shows how smoothing constraints translate to real signal bandwidth limitations using an FFT analysis.
>
> >  "Since the RL algorithm relies on a full simulation of the system, it is hard in its current form to scale up, i.e. to more than 10 qubits."
>
> A1.4: We agree with this observation. However, one could replace a full system simulation with an ML based emulator, or direct physical device sampling [6]. Given such improvements, our proposed method could scale to larger systems. We also want to highlight that control of smaller dimensional systems is an important and relevant research direction for advancing quantum technologies.
>
> > Minor comment
>
> Thanks for spotting this, the third term intentionally has the same structure but there is a typo $S(\Omega_i)$ should read $S(\Delta_i)$.
>
> ### References:
>
> [1] Rigetti Computing. (2025). Pulses and waveforms. pyQuil 4.16.1. https://pyquil-docs.rigetti.com/en/stable/quilt_waveforms.html
>
> [2] IQM. (2025). IQM Pulla 6.15. https://docs.meetiqm.com/iqm-pulla/
>
> [3] Magnard, P., et. al. Physical Review Letters, 121(6), 060502.
>
> [4] Willsch, Dennis. arXiv preprint arXiv:2008.13490 (2020).
>
> [5] Bhatnagar, S., Lakshmanan, K. J Optim Theory Appl 153, 688–708 (2012).
>
> [6] Baum, Yuval, et al. PRX Quantum, vol. 2, no. 4, 2021, p. 040324.

---

> > ### Comment · Reviewer_sD3o · 2025-04-02
> >
> > I thank the authors for the response. I have carefully read the other reviews and authors' responses. I believe most issues pointed by reviewers are addressed adequately. My score remains the same.
> >
> > The other reviews remind me of a paper [1] that is missing but worth mentioning in related work, where they use an RL framework to find control schemes for a transmon-cavity system for error correcting.
> >
> > [1] Sivak, Volodymyr V., et al. "Real-time quantum error correction beyond break-even." Nature 616.7955 (2023): 50-55.

---

> > > ### Author Response · Authors · 2025-04-06
> > >
> > > > The other reviews remind me of a paper [1] that is missing but worth mentioning in related work, where they use an RL framework to find control schemes for a transmon-cavity system for error correcting.
> > >
> > > We thank the reviewer for the comment and for highlighting this paper. It is related to some of the papers mentioned by reviewer Vwkd, and optimises parameters at the circuit level, which differs from our work which addresses pulse level control; the real time feedback aspect is related to the results discussed Sec. 5.4. We now discuss this paper in the related work section. Furthermore, in the conclusion, we state that extending our work to several concatenated pulses (i.e. circuit level control) constitutes an interesting direction for future work.

---

### Decision · Program_Chairs · 2025-05-01

**Decision:**

Accept (poster)

**Comment:**

The rebuttal addressed the concerns from the reviewers and provided comprehensive feedback. It is useful for assessing the paper's contributions. Most reviewers recommend acceptance after discussion, and the ACs concur. The final version should include all reviewer comments, suggestions, and additional discussion from the rebuttal.